



# OpenFOAM-avalanche 2312: Depth-integrated Models Beyond Dense Flow Avalanches

Matthias Rauter[1,2] and Julia Kowalski[3]

[1]Unit of Geotechnical Engineering, University of Innsbruck, Innsbruck, Austria
[2]Department of Civil Engineering and Natural Hazards, University of Natural Resources and Life Sciences, Vienna, Austria
[3]Methods for Model-based Development in Computational Engineering, RWTH Aachen University, Aachen, Germany

**Correspondence:** Julia Kowalski (kowalski@mbd.rwth-aachen.de)

**Abstract.** Numerical simulations have become an important tool for the estimation and mitigation of gravitational mass flows, such as avalanches, landslides, pyroclastic flows or turbidity currents. Depth-integration stands as a pivotal concept in rendering numerical models applicable to real-world scenarios, as it provides the required efficiency and a streamlined workflow for geographic information systems. In recent years, a large number of flow models were developed following the idea of depth-integration, thereby enlarging the applicability and reliability of this family of process models substantially. It has been previously shown that the Finite Area Method of OpenFOAM® can be utilized to express and solve the basic depth-integrated models representing incompressible dense flows. In this manuscript, the previous work (Rauter et al., 2018) is extended beyond the dense flow regime to account for suspended particle flows, such as turbidity currents and powder snow avalanches. A novel coupling mechanism is introduced to enhance the simulation capabilities for mixed snow avalanches. Further, we will give an updated description of the revised computational framework, its integration into OpenFOAM and interfaces to geographic information systems. This work aims to provide practitioners and scientists with an open source tool that facilitates transparency and reproducibility and that can be easily applied to real world scenarios. The tool can be used as a baseline for further developments and in particular allows for modular integration of customized process models.

## 1 Introduction

Run-out and impact simulations of gravitational mass flows typically rely on depth-integrated models (e.g. Pitman et al., 2003; Sampl and Zwinger, 2004; Christen et al., 2010; Iverson and George, 2014; Mergili et al., 2017). In comparison with fully resolved three-dimensional models, this framework provides a range of upsides: The computational expense is substantially reduced, interface and phase tracking are simpler and more reliable, integration in geographic information systems is straight-forward. The model is easier to solve numerically, to set up, to calibrate and to evaluate. However, depth-integration comes at a price: The vertical flow structure including the shear gradient is lost and all related effects, if needed for heuristic closures, have to be reintroduced with empirical models. This includes friction, erosion of basal material and its deposition (e.g. Rauter and Köhler, 2020), as well as layering of varying regimes (e.g. Bartelt et al., 2016). A possibility to overcome this is the shallow moment approach (Kowalski and Torrilhon, 2019), however, which has not been applied successfully to real-scenario granular mass flows yet. Nevertheless, depth-integrated models have proven to be a good compromise between simplicity and





complexity, especially for flows of geographic extended from avalanches (Christen et al., 2010) to tsunamis (Løvholt et al.,
2015).

Granular flows show a large variety of behaviours. A very strong distinction of properties can be linked to the Stokes number
$St$, expressing the ratio between inertia and drag forces on particles (Boyer et al., 2011; Rauter, 2021). For a flow with shear
rate $\dot{\gamma}$ of granules with density $\rho_{\mathrm{g}}$ and diameter $d$, in a medium of viscosity $\nu_{\mathrm{c}}$ and density $\rho_{\mathrm{c}}$, the Stokes number can be written
as

$$St = d^2 \frac{\dot{\gamma}\,\rho_{\mathrm{g}}}{\nu_{\mathrm{c}}\,\rho_{\mathrm{c}}}. \tag{1}$$

At high Stokes numbers, drag forces are small and particles move freely through the surrounding fluid or gas. Thus the
bulk motion is dominated by particle-particle interactions and particles will arrange in a well defined and relatively high
packing density that only depends on the local shear rate and pressure (e.g. Forterre and Pouliquen, 2008). Furthermore, for
many realistic problems, the bulk density can be assumed constant with acceptable accuracy. Dense flow models often take
advantage of this fact and are formulated as incompressible non-Newtonian fluids (e.g. Savage and Hutter, 1989; Rauter, 2021).

At low Stokes numbers, drag on particles is substantial and particles are not able to rearrange freely within the carrier
medium. Particles and surrounding fluid form a suspension and move like a single fluid, only to be slowly separated by the
settling velocity. The packing density depends on various aspects and most importantly on the history of the flow. This is a
strong hint that the packing density requires an evolution equation to be properly described (as done by e.g. Parker et al., 1986;
Kowalski and McElwaine, 2013; Bartelt et al., 2016; Issler et al., 2018; Rauter, 2021).

It can be seen from Eq. (1) that the Stokes number depends on the particle size. In polydisperse granular flows, i.e. flows
with particles of various sizes (e.g. Barker et al., 2021), this can lead to vertical segregation of small and large particles and
thus a coexistence of both regimes. This can be well observed in snow avalanches (Sovilla et al., 2015), where a dense flow
is formed by relatively coarse snow blocks of size $10^{-2}\,\mathrm{m}$ (Rauter et al., 2018) and a powder cloud is formed by small ice
particles of size $10^{-4}\,\mathrm{m}$ (Rastello et al., 2011; Bartelt et al., 2016), see Fig. 1.

In terms of depth-integrated models this calls for a two-layer model, capturing the dense flow with an incompressible model
and the powder cloud with a suspension model (Sampl and Zwinger, 2004; Bartelt et al., 2016).

In this work, we will extend the dense flow model of Rauter et al. (2018) to low Stokes number suspension flows following
the model of Parker et al. (1986). We will make and evaluate some adjustments to account for high density differences between
the carrier medium and the particles. In a further step, we will combine the models for dense flow and suspension into a two-
layer model, capable of simulating mixed snow avalanches, similar to Turnbull and Bartelt (2003) and Bartelt et al. (2016). For
this purpose, we have to define a coupling mechanism, i.e. a mass flux term that feeds the powder cloud from the dense core.
We develop a novel idealized relation, that encapsulates the essential relations of this process and deliberately avoids more
complex mechanisms (e.g. Sampl and Zwinger, 2004; Bartelt et al., 2016). We focus on clarity, simplicity and modularity,
and therefore describe all processes with simple, local relations that can be formulated independently of one another. This
is motivated by the goal to get a simple baseline model but also by the observation that complexity not necessarily leads to
better results (Zhao and Kowalski, 2022). The natural terrain is handled as described previously by Rauter et al. (2018). While





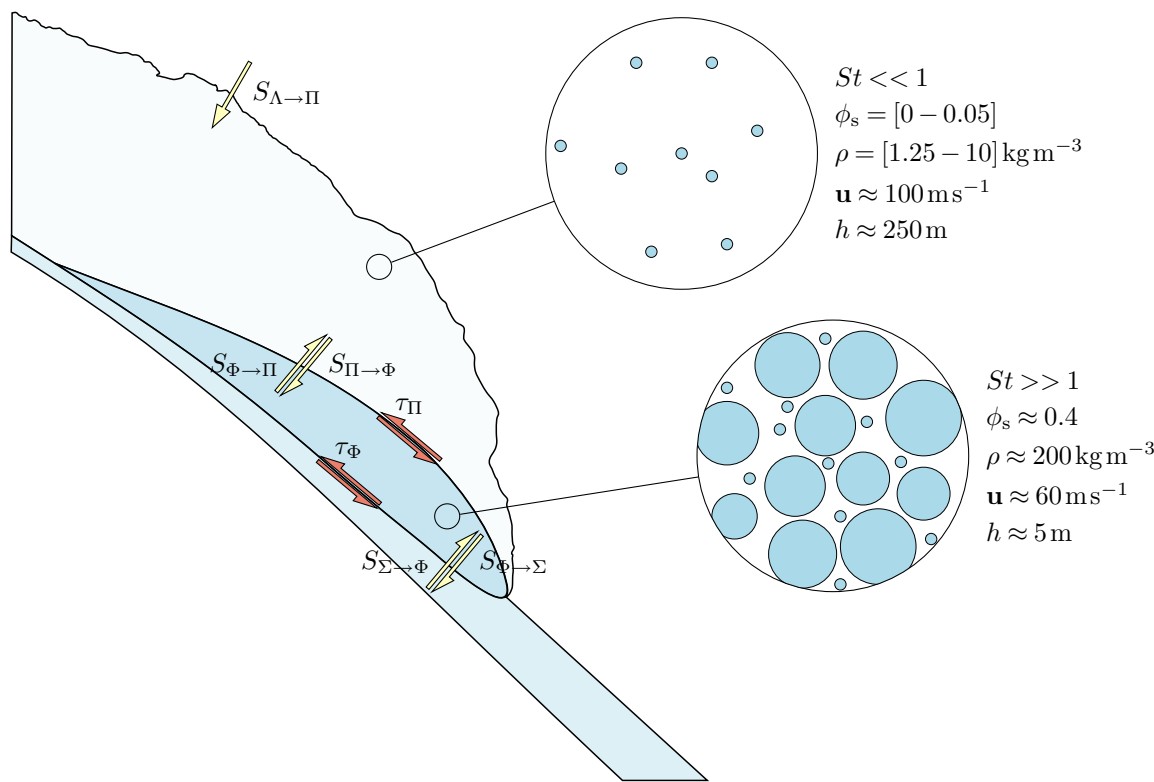

**Figure 1.** Conceptual sketch of a mixed powder snow avalanche, combining an incompressible dense flow of high Stokes number with a variable density suspension cloud characterized by a small Stokes number. The avalanche growth is controlled by the erosion of the intact snow cover and the entrainment of ambient air, the layers are interacting through mass (yellow) and momentum fluxes (red). Characteristic scales of packing density $\phi_\mathrm{s}$, bulk density $\rho$, velocity $\mathbf{u}$ and height $h$ vary substantially between layers and thus require individual models.

the main focus of the presented work is snow avalanches, the implementation might very well be useful for the simulation of
turbidity currents, as several researchers suspect a dense core in these flows as well (e.g. Heerema et al., 2020).

The naming convention of layers and fluxes follows Bartelt et al. (2016), the dense core is denoted with $\Phi$, the suspension flow with $\Pi$, the static bottom layer with $\Sigma$ and the stationary ambient fluid with $\Lambda$. Flow fields are marked with the respective subscripts and fluxes between layers with two subscripts and an arrow indicating the direction of the flux (see Fig. 1).

The numerical solution and implementation are based on the Finite Area Method (Tuković and Jasak, 2012; Rauter and
Tuković, 2018) as implemented in OpenFOAM. Its modular structure and building blocks have proven to be flexible and highly valuable for physical depth-integrated models. Various code parts are reused between all models and various communities, in particular the numerical solver, geometry and data handling but also various physical code, such as friction models. Beside the introduction of the new model and capabilities, this work should highlight the capability of extending the basic OpenFOAM solver to complex models.



The toolchain to process the basic terrain data, all the way to the final simulation visualisation was improved substantially since the work of Rauter et al. (2018) and many external dependencies were removed, in order to facilitate a tight integration into OpenFOAM. As such, this paper also represents an updated description of the toolchain and practical applications. In this context we will also give a revised introduction into the Finite Area Method and the specific derivations of depth-integration. The model is aimed equally at practitioners, providing a simple mixed snow avalanche model but also to scientists, providing

an open model and framework that can be easily modified and extended to evaluate new concepts and ideas.

     The novel model is evaluated with various synthetic test cases and finally applied to two real scale events, namely the 1988 Wolfsgruben avalanche and the 2019 Eiskar avalache.

## 2   Foundation and Framework

### 2.1   Conservation Equations and Depth-integration

The presented method fundamentally relies on balance equations, in particular, the conservation of mass and momentum for fluids. The combination of these two equations is widely known as Navier-Stokes Equations (e.g. Ferziger and Peric, 2002) and can be written as

$$\frac{\partial \rho}{\partial t} + \boldsymbol{\nabla} \cdot (\rho \, \mathbf{u}) = 0, \tag{2}$$

$$\frac{\partial \rho \, \mathbf{u}}{\partial t} + \boldsymbol{\nabla} \cdot (\rho \, \mathbf{u} \, \mathbf{u}) = \boldsymbol{\nabla} \cdot \mathbf{T} + \mathbf{f}, \tag{3}$$

with the bulk density $\rho$ and the bulk velocity $\mathbf{u}$. (Note that it can also be defined for an individual phase with some modifications, see, e.g. Rauter, 2021). These flow fields are functions of time $t$ and space $\mathbf{x} = (x, y, z)^T$. The model (2) and (3) describes their evolution from a known state $\mathbf{u}(0, \mathbf{x}) = \mathbf{u}_0$, $\rho(0, \mathbf{x}) = \rho_0$, (initial condition) under the influence of boundary conditions. The divergence of the stress tensor $\mathbf{T}$ acts as diffusion of momentum, the volume force $\mathbf{f}$ represents additional forces, such as gravitational acceleration.

Appropriate closure relations that express the stress tensor $\mathbf{T}$ as a function of the unknown flow fields yield a well-posed problem that can, in principle, be solved with numerical methods (Barker and Gray, 2017). However, even a well-posed problem is often not practically feasible from a computational perspective. Therefore, multiple simplifications have to be made to make problems of practical relevance accessible. Simplifications often come in the form of averaging over a certain time or over space to get rid of turbulent structures (Reynolds-averaging, see e.g. Ferziger and Peric, 2002), to describe the average

behaviour of multiple interpenetrating phases (phase-averaging, e.g. Rauter, 2021) or to get rid of the vertical dimension (e.g. Savage and Hutter, 1989; Rauter and Tuković, 2018). The latter is referred to as depth-averaging or depth-integration and avoids the calculation of three dimensional flow details. It yields mean values of e.g. density $\overline{\rho}$ and velocity $\overline{\mathbf{u}}$ along the depth.





In the simplest case, where the depth-integration is aligned with a spatial axis, e.g. the z-axis, the problem can be reduced from three $(x, y, z)$ to two dimensions $(x, y)$. In this case, the depth-averaged value for an arbitrary field $\psi$ is defined as

$$\overline{\psi}(x, y, t) = \frac{1}{h} \int_0^h \psi(x, y, z, t)\, \mathrm{d}z \tag{4}$$

The newly introduced field $h(x, y, t)$ describes the flow depth, here in terms of the z-coordinate of the top boundary of the integration, for a bottom boundary assumed to be aligned with $z = 0$. The bottom and top boundaries are usually defined such that the mass flux through them is zero, meaning that they move with the vertical velocity of the flow at the respective position. The simplest example of such a model are the Shallow Water Equations (Barré de Saint-Venant, 1871). Defining the boundary in any other way, will lead to additional source or sink terms, depending on the mass flux through the boundary (Pudasaini and Hutter, 2007). Examples would be any kind of entrainment and deposition fluxes.

Depth-integrated models are often considered synonymous with two-dimensional models. However, real avalanches and landslides travel along paths and surfaces in three-dimensional space. The three-dimensional nature of the terrain has to be reintroduced by modifying the two-dimensional model equations. Most often this is accomplished by abandoning Cartesian coordinate systems and Euclidean geometry, which was described in detail first by Savage and Hutter (1989, 1991) and many others in more detail and accuracy since then (e.g. Bouchut and Westdickenberg, 2004; Denlinger and Iverson, 2004; Pudasaini et al., 2005; Hergarten and Robl, 2015). This introduces various correction terms based on Christoffel formalism that are difficult to handle in complex models. In practice, idealized approximations are frequently employed (e.g. in RAMMS, see Fischer et al., 2012), leading to a disparity between theory and practical implementation.

An alternative to two-dimensional models with excessive curvature terms is the direct solution of the governing equations in three-dimensional space (Craster and Matar, 2009; Hagemeier et al., 2011; Rauter and Tuković, 2018). Depth-integration is still compatible with this approach and it can in principle be conducted in any direction pointing out of the surface. Yet in this work, depth-integration is always conducted in direction of the normal vector $\mathbf{n}^\Gamma$ to the flow surface $\Gamma$, as shown in Fig. 2. This has formally to be conducted in a surface aligned coordinate system $x'$-$y'$-$z'$,

$$\overline{\psi}(\mathbf{x}_\mathrm{b}) = \frac{1}{h} \int_0^h \det(\mathbf{J})\, \psi\left(\mathbf{x}', t\right)\, \mathrm{d}z' \approx \frac{1}{h} \int_0^h \psi\left(\mathbf{x}', t\right)\, \mathrm{d}z'. \tag{5}$$

The Jacobi-matrix $\mathbf{J}$, representing the transformation $\partial \mathbf{x}'/\partial \mathbf{x}$ and its determinant $\det(\mathbf{J})$ take into account the curvature of the surface and its influence of the volume in a differential volume element of the flow (Bouchut et al., 2003). This effect is of order $h/R$ (Bouchut et al., 2003) with the mean curvature radius $R$, and thus small for mildly curved surfaces ($R$ is small in comparison to the flow height $h$). As in most other models, the influence of the curvature on depth-integration is ignored in this work.

## 2.2 Surface Partial Differential Equations

Depth-integration in terms of Eq. (5) projects all three-dimensional flow fields onto the surface $\Gamma$ they are constrained by. The conservation equations can then be expressed as surface partial differential equations (SPDEs) that are defined on the surface





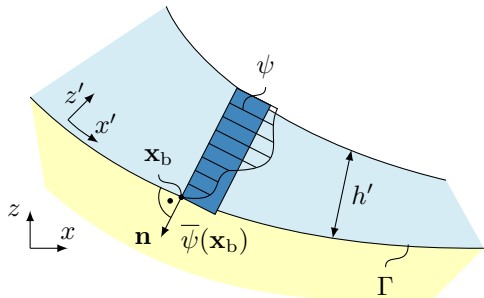

**Figure 2.** Depth integration reduces the full three-dimensional flow field $\psi$ (dashed area) to an average flow field $\overline{\psi}$ (blue filled area), that is assigned to a point $\mathbf{x}_\mathrm{b} \in \Gamma$.

$\Gamma$ and include derivatives of various fields along it. These derivatives are emerging from depth-integrating the ordinary three-dimensional Nabla operator $\boldsymbol{\nabla}$. The Nabla operator is a vector of the derivatives in all directions and can be expressed in terms of Cartesian coordinates as

$$\boldsymbol{\nabla} = \left( \frac{\partial}{\partial x}, \frac{\partial}{\partial y}, \frac{\partial}{\partial z} \right)^T. \tag{6}$$

The directional derivative of an arbitrary field $\psi(\mathbf{x})$, e.g. in direction of the flow surface normal $\mathbf{n}^\Gamma$ can be calculated with the scalar product,

$$\frac{\partial \psi}{\partial \mathbf{n}^\Gamma} = \boldsymbol{\nabla} \psi \cdot \mathbf{n}^\Gamma. \tag{7}$$

The surface tangential derivative is consequently obtained by subtracting the derivative in normal direction of the surface (including the respective direction, $\mathbf{n}^\Gamma$)

$$\boldsymbol{\nabla}_\mathrm{s}^\Gamma \psi = \boldsymbol{\nabla} \psi - \left( \boldsymbol{\nabla} \psi \cdot \mathbf{n}^\Gamma \right) \mathbf{n}^\Gamma = \left( \mathbf{I} - \mathbf{n}^\Gamma \mathbf{n}^\Gamma \right) \cdot \boldsymbol{\nabla} \psi, \tag{8}$$

with the identity matrix $\mathbf{I}$ (multiplications of vectors without dot express the outer product $n_i\, n_j$). The matrix $\mathbf{P} := \mathbf{I} - \mathbf{n}^\Gamma \mathbf{n}^\Gamma$ hence defines a projection matrix that maps a vector $\boldsymbol{\psi}$ to the surface's tangential space and constitutes the surface tangential gradient operator (Deckelnick et al., 2005). Per its local definition, the surface tangential derivative does not incorporate local curvature information. The surface gradient with respect to the complex surface topography, however, acknowledges local curvature, and can be written as

$$\boldsymbol{\nabla}^\Gamma \psi = \boldsymbol{\nabla}_\mathrm{s}^\Gamma \psi + \kappa\, \psi\, \mathbf{n}^\Gamma, \tag{9}$$

in which $\kappa$ denotes the local Gaussian curvature (Dieter-Kissling et al., 2015; Tuković and Jasak, 2012). Since $\boldsymbol{\nabla}_\mathrm{s}^\Gamma$ projects on the tangential space, it does not contain normal components. As a consequence, the normal directed contribution to $\boldsymbol{\nabla}^\Gamma \psi$ is solely determined by the curvature term $\kappa\, \psi\, \mathbf{n}^\Gamma$. By defining $\boldsymbol{\nabla}_\mathrm{n}^\Gamma \psi = \kappa\, \psi\, \mathbf{n}^\Gamma (= \mathbf{n}^\Gamma \mathbf{n}^\Gamma \boldsymbol{\nabla}^\Gamma \psi)$ the decomposition into tangential and normal direction reads

$$\boldsymbol{\nabla}^\Gamma \psi = \nabla_\mathrm{s}^\Gamma \psi + \nabla_\mathrm{n}^\Gamma \psi. \tag{10}$$



Following the same rationale, we can compute the surface tangential derivative as

$$\boldsymbol{\nabla}_{\mathrm{s}}^{\Gamma} = \boldsymbol{\nabla}^{\Gamma} \cdot \left( \mathbf{I} - \mathbf{n}^{\Gamma} \mathbf{n}^{\Gamma} \right),\tag{11}$$

and all curvature effects as

$$\boldsymbol{\nabla}_{\mathrm{n}}^{\Gamma} = \boldsymbol{\nabla}^{\Gamma} \cdot \left( \mathbf{n}^{\Gamma} \mathbf{n}^{\Gamma} \right) = \kappa \, \mathbf{n}^{\Gamma},\tag{12}$$

without the requirement to explicitly calculate the curvature. The surface gradient can be easily calculated with Gauss Surface

Theorem (Dieter-Kissling et al., 2015; Tuković and Jasak, 2012; Rauter and Tuković, 2018).

It remains to be established how the differential operators from three-dimensional models can be depth-integrated. They are, similar to ordinary fields, see Eq. (5), integrated in surface normal direction and in the surface aligned coordinate system $x'$-$y'$-$z'$,

$$
\begin{aligned}
\overline{\boldsymbol{\nabla} \psi}(\mathbf{x}_{\mathrm{b}}) &= \frac{1}{h} \int_{0}^{h} \det(\mathbf{J}) \, \boldsymbol{\nabla}' \psi(\mathbf{x}') \, \mathrm{d}z' \approx \frac{1}{h} \int_{0}^{h} \boldsymbol{\nabla}' \psi(\mathbf{x}') \, \mathrm{d}z' \\
&= \frac{1}{h} \int_{0}^{h} \left( \boldsymbol{\nabla}^{\Gamma} \psi(\mathbf{x}') + \frac{\partial}{\partial z'} \psi(\mathbf{x}') \, \mathbf{e}'_{z} \right) \mathrm{d}z' \\
&= \frac{1}{h} \int_{0}^{h} \boldsymbol{\nabla}^{\Gamma} \psi(\mathbf{x}') \, \mathrm{d}z' + \frac{1}{h} \int_{0}^{h} \frac{\partial}{\partial z'} \psi(\mathbf{x}') \, \mathrm{d}z' \, \mathbf{e}'_{z} \\
&= \int_{0}^{h} \frac{\boldsymbol{\nabla}^{\Gamma} \psi(\mathbf{x}')}{h} \, \mathrm{d}z' + \frac{\psi(\mathbf{x}_{\mathrm{t}}) - \psi(\mathbf{x}_{\mathrm{b}})}{h} \mathbf{n}^{\Gamma} \\
&= \boldsymbol{\nabla}^{\Gamma} \overline{\psi}(\mathbf{x}_{\mathrm{b}}) + \frac{\psi(\mathbf{x}_{\mathrm{t}}) - \psi(\mathbf{x}_{\mathrm{b}})}{h} \mathbf{n}^{\Gamma},
\end{aligned}
\tag{13}
$$

where $\mathbf{x}_{\mathrm{b}}$ is a point on the bottom of the flow (and thus the flow surface $\Gamma$) and $\mathbf{x}_{\mathrm{t}}$ the corresponding point on the free surface

of the flow. The second term on the right hand side of Eq. (13) represents an additional sink or source term, that arises if $\psi$ is not zero at the bottom, $\mathbf{x}_{\mathrm{b}}$, or the top of the flow, $\mathbf{x}_{\mathrm{t}}$, for example entrainment or basal friction.

Due to the depth-integration in a surface aligned coordinate system, the surface derivative $\boldsymbol{\nabla}^{\Gamma}$ appears in the depth-integrated conservation equations. It is decomposed into surface normal and surface tangential components with Eqs. (11) and (12). In the momentum conservation equation, tangential components determine the velocity evolution while surface normal components

determine the basal pressure. In addition to surface normal components that emerge from curvature effects, there will be normal components appearing due to local sources, such as the gravitational acceleration. In the classical Shallow Water Equations and the Savage and Hutter (1989, 1991) model, this partition is simply achieved by separating the conservation equations in x- and y-direction from the equation in z-direction.

With these building blocks, and some knowledge on how to transform one-dimensional shallow flow models (e.g. Savage

and Hutter, 1989; Parker et al., 1986), it is possible to extend nearly arbitrary depth-integrated flow models to complex terrain.



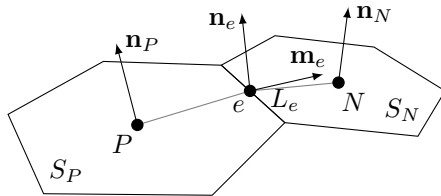

**Figure 3.** A finite area cell $P$ and its neighbour $N$, used to calculate the approximation of surface derivatives in terms of the surface Gauss theorem, integrating fluxes through cell edges e with length $L_\mathrm{e}$ and outward pointing vector $\mathbf{m}_\mathrm{e}$.

In particular, ordinary depth-integrated flow models represent the surface tangential momentum conservation equation and the flow depth equation. The two-dimensional $\boldsymbol{\nabla}$-operators have to be replaced with the surface tangential $\boldsymbol{\nabla}_\mathrm{s}^\Gamma$-operators. The surface normal momentum conservation equation can be applied to replace the usually simplified expression for the basal pressure.

## 2.3 Finite Area Method

Partial Differential Equations, as well as their SPDE counterparts, are rarely solvable in an analytical sense, especially practical problems that represent real world situations (Ferziger and Peric, 2002). Therefore, we rely on numerical approximations of SPDEs and the Finite Area Method. This method is a variation of the Finite Volume Method (see Ferziger and Peric, 2002; Jasak, 1996; Moukalled et al., 2016, for details) in $N + 1$ dimensions, where $N$ is the dimension of the control volumes. This means that for two-dimensional control volumes (i.e. surfaces), vectorial entities, such as normal vectors, velocities or fluxes, will be three-dimensional. Similar to the conventional Finite Volume Method, the Gaussian Surface Theorem (Tuković and Jasak, 2012) is applied and discretized by simplifying a control surface $S$ as a flat, convex polygon $S_i$, as shown in Fig. 3. The expressions for the differential operators follow as

$$\boldsymbol{\nabla}^\Gamma \psi = \frac{1}{S} \oint_{\partial S} \mathbf{m}^\Gamma \psi \, \mathrm{d}L \approx \frac{1}{S_i} \sum \psi_e \, \mathbf{m}_e \, L_e \tag{14}$$

and

$$\boldsymbol{\nabla}^\Gamma \cdot \boldsymbol{\psi} = \frac{1}{S} \oint_{\partial S} \mathbf{m}^\Gamma \cdot \boldsymbol{\psi} \, \mathrm{d}L \approx \frac{1}{S_i} \sum \boldsymbol{\psi}_e \cdot \mathbf{m}_e \, L_e. \tag{15}$$

Index $e$ refers to a discrete number of straight edges that form the polygon with surface $S_i$. $\psi_e$ is the average value of the field $\psi$ on the edge $e$, $L_e$ its length and $\mathbf{m}_e$ the $\Gamma$-tangential and edge-normal outward pointing vector. $S_i$, $L_e$ and $\mathbf{m}_e$ are purely geometrical properties that are defined during mesh generation. Values of fields on edges $\psi_e$, on the other hand, are interpolated from values of edge-adjacent cells, $\psi_P$ and $\psi_N$. This introduces flux transport across cells and represents the flow of mass or information from one cell to neighbouring ones. The fluxes can then be associated in a linear system of equations that is solved with a suitable method.





Discretization of non-gradient terms, e.g. the temporal derivative or any source term, is done in complete analogy to the Finite Volume Method and obtained from integration over the control surface $S_i$. For details we refer to the large amount of

excellent literature on the Finite Volume Method (Ferziger and Peric, 2002; Jasak, 1996; Moukalled et al., 2016; LeVeque, 2002).

## 3   Dense Flow Model

The dense flow model describes the flow of incompressible material with density $\rho_\Phi$ (see Fig. 1). In case of a granular mass flow, the density follows from the grain density $\rho_{\mathrm{g}}$ and the volumetric packing density $\phi_\Phi$ as

$$\rho_\Phi = \phi_\Phi \, \rho_{\mathrm{g}}. \tag{16}$$

However, fluids can be simulated with this model as well, in which case $\rho_\Phi$ is the intrinsic density of the fluid. The depth-integrated mass and momentum conservation equations follow as

$$\frac{\partial h_\Phi}{\partial t} + \boldsymbol{\nabla}^\Gamma \cdot \left(h_\Phi \, \overline{\mathbf{u}}_\Phi\right) = \frac{S_\Phi^\phi}{\phi_\Phi}, \tag{17}$$

$$\frac{\partial h_\Phi \, \overline{\mathbf{u}}_\Phi}{\partial t} + \xi_\Phi \, \boldsymbol{\nabla}_{\mathrm{s}}^\Gamma \cdot \left(h_\Phi \, \overline{\mathbf{u}}_\Phi \, \overline{\mathbf{u}}_\Phi\right) = -\frac{\boldsymbol{\tau}_\Phi}{\rho_\Phi} + h_\Phi \, \mathbf{g}_{\mathrm{s}} - \frac{1}{2 \, \rho_\Phi} \, \boldsymbol{\nabla}_{\mathrm{s}}^\Gamma \left(h_\Phi \, p_\Phi\right) + \frac{S_\Phi^{\mathrm{u}}}{\rho_\Phi}, \tag{18}$$

$$\xi_\Phi \, \boldsymbol{\nabla}_{\mathrm{n}}^\Gamma \cdot \left(h_\Phi \, \overline{\mathbf{u}}_\Phi \, \overline{\mathbf{u}}_\Phi\right) = h_\Phi \, \mathbf{g}_{\mathrm{n}} - \frac{1}{2 \, \rho_\Phi} \, \boldsymbol{\nabla}_{\mathrm{n}}^\Gamma \left(h_\Phi \, p_\Phi\right) - \frac{1}{\rho_\Phi} \, \mathbf{n}^\Gamma \, p_\Phi. \tag{19}$$

The unknown flow fields are the flow depth $h_\Phi$, the depth-integrated velocity $\overline{\mathbf{u}}_\Phi$ and the basal pressure $p_\Phi$. The gravitational acceleration is represented by its surface tangential projection $\mathbf{g}_{\mathrm{s}} = \left(\mathbf{I} - \mathbf{n}^\Gamma \, \mathbf{n}^\Gamma\right) \mathbf{g}$ and its surface normal projection $\mathbf{g}_{\mathrm{n}} = \left(\mathbf{n}^\Gamma \, \mathbf{n}^\Gamma\right) \mathbf{g}$. Equation (19) represents the surface normal component of the momentum conservation equation and yields the basal pressure $p_\Phi$.

The factor $\xi_\Phi$ denotes the shape factor that compensates for errors introduced by switching integration and multiplication, namely $\xi_\Phi \, \overline{\mathbf{u}}_\Phi \, \overline{\mathbf{u}}_\Phi = \overline{\mathbf{u}_\Phi \, \mathbf{u}_\Phi}$. It depends on the velocity profile and as such on the constitutive model and the state of the flow. It is usually neglected or set to a theoretical and constant value, derived e.g. from the Bagnold (1954) velocity profile ($\xi_\Phi = 5/4$).

### 3.1   Friction in the Dense Flow Model

The term $\boldsymbol{\tau}_\Phi$ represents the depth-integrated divergence of the shear stress tensor and thus the constitutive model of the flowing

mass. Assuming that the top boundary is stress free and that surface tangential derivatives of the deviatoric stress tensor are small, the only remaining entity is the basal friction. In this work, we will use the friction model presented by Rauter et al. (2016), which is closely related to the widely used Voellmy (1955) friction model. It is given as

$$\boldsymbol{\tau}_\Phi = \left(\mu \, p_\Phi + \frac{\rho_\Phi \, |\mathbf{g}|}{\chi \, h_\Phi^2} \, |\overline{\mathbf{u}}_\Phi|^2\right) \frac{\overline{\mathbf{u}}_\Phi}{|\overline{\mathbf{u}}_\Phi|}, \tag{20}$$

with dry friction coefficient $\mu$ and turbulent friction coefficient $\chi$. A wide range of alternative friction models can be found in

the literature and a number of them are implemented into the presented software.





## 3.2 Entrainment and Deposition in the Dense Flow Model

$S_\Phi^\phi$ represents the sum of all volumetric source and sink terms of grains, e.g. erosion and entrainment of additional mass or its deposition, $S_\Phi^{\mathrm{u}}$ represents its associated momentum. Dividing by the packing density in Eq. (17) simplifies handling of density changes in the different flow regimes. In the simplest case, e.g. laboratory experiments on inclined planes or chutes, the source and sink terms are zero.

For snow avalanches and many other realistic gravitational mass flows, entrainment of erodible material along the avalanche path plays an important role. A popular entrainment model can be derived by comparing the dissipated energy in the mass flow with the energy required to mobilize the static material (Fischer et al., 2015),

$$S_{\Sigma \to \Phi}^\phi = \frac{\boldsymbol{\tau}_\Phi \cdot \overline{\mathbf{u}}_\Phi}{\rho_\Phi \, e_\mathrm{b}} \, \phi_\Phi, \tag{21}$$

with the specific erosion energy $e_\mathrm{b}$ as the single parameter. Here it is assumed that the packing density of the static layer is the same as in the dense flow $\phi_\Phi$.

Rauter and Köhler (2020) presented an extension to account for the deposition of flowing material, $S_{\Phi \to \Sigma}^\phi$. This aspect is neglected in this work and the flow height of the last time step is assumed to be the final deposition of the model.

The total flux term between the static layer and the flowing avalanche is determined as the difference between entrainment and deposition,

$$S_\Phi^\phi = S_{\Sigma \to \Phi}^\phi - S_{\Phi \to \Sigma}^\phi. \tag{22}$$

The related momentum source and sink terms are zero in the case of single layer flows, as both erodible and deposited material is static.

The height (in surface normal direction) of the static material on the topography can be tracked with an additional evolution equation,

$$\frac{\partial h_\Sigma}{\partial t} = \frac{S_\Sigma^\phi}{\phi_\Phi}, \tag{23}$$

with

$$S_\Sigma^\phi = S_{\Phi \to \Sigma}^\phi - S_{\Sigma \to \Phi}^\phi, \tag{24}$$

again under the assumption that the static layer has the same packing density as the flowing avalanche $\phi_\Phi$. Tracking the thickness of the static layer allows to limit the available entrainable material, hence to turn of entrainment if the erodible layer is depleted.

## 4 Suspension Flow Model

The suspension flow model describes the flow of a dynamic mixture of a granular material of density $\rho_\mathrm{g}$ and the surrounding fluid of density $\rho_\mathrm{c}$. It corresponds, to some degree, to a depth-integration of the compressible model of Rauter (2021). The





mixture density follows as

$$\rho_\Pi = \phi_\Pi\,\rho_{\mathrm{g}} + (1 - \phi_\Pi)\,\rho_{\mathrm{c}}, \tag{25}$$

with the variable packing density or phase fraction $\phi_\Pi$. Introducing the buoyancy density ratio,

$$r = \frac{\rho_{\mathrm{g}} - \rho_{\mathrm{c}}}{\rho_{\mathrm{c}}}, \tag{26}$$

the mixture density can be expressed as

$$\rho_\Pi = \rho_{\mathrm{c}}\,(1 + \phi_\Pi\,r). \tag{27}$$

The buoyancy assumption, an often applied simplification (e.g. Parker et al., 1986), implies that $\phi_\Pi \lessapprox 10^{-2}$ and $r \approx 1$ and thus $\rho_\Pi \approx \rho_{\mathrm{c}}$. This is reasonable if $\rho_{\mathrm{g}}$ and $\rho_{\mathrm{c}}$ are at least similar in order of magnitude, e.g. sand in water. However, this does not hold for snow avalanches, i.e. mixtures of grains or ice ($\rho_{\mathrm{g}} \approx 1000\,\mathrm{kg\,m^{-3}}$) with air ($\rho_{\mathrm{c}} \approx 1\,\mathrm{kg\,m^{-3}}$). Thus, we will omit this assumption and consider the dynamic density as given by Eq. (27) in all terms.

Due to the variable mixture, there will be two phases that have to be described by balance laws. In depth-averaged frameworks, this is usually handled by describing the total volume occupied by the flowing masses (grains and flowing ambient fluid) in terms of the flow depth $h_\Pi$ and the volume of grains, expressed by the depth-integrated volume fraction $h_\Pi\,\overline{\phi}_\Pi$ (Parker et al., 1986; Bartelt et al., 2016; Kowalski and McElwaine, 2013). The phases are assumed to move with the same velocity $\overline{\mathbf{u}}_\Pi$, differences in velocity (e.g. settling of particles) are considered with empirical corrections.

The depth-integrated mass and momentum conservation equations follow as

$$\frac{\partial h_\Pi}{\partial t} + \boldsymbol{\nabla}^\Gamma \cdot (h_\Pi\,\overline{\mathbf{u}}_\Pi) = S_\Pi^{\mathrm{h}}, \tag{28}$$

$$\frac{\partial \overline{\phi}_\Pi\,h_\Pi}{\partial t} + \boldsymbol{\nabla}^\Gamma \cdot \left(\overline{\phi}_\Pi\,h_\Pi\,\overline{\mathbf{u}}_\Pi\right) = S_\Pi^{\phi}, \tag{29}$$

$$\frac{\partial\left(1 + r\,\overline{\phi}_\Pi\right)h_\Pi\,\overline{\mathbf{u}}_\Pi}{\partial t} + \xi_\Pi\,\boldsymbol{\nabla}_{\mathrm{s}}^\Gamma \cdot \left(\left(1 + r\,\overline{\phi}_\Pi\right)h_\Pi\,\overline{\mathbf{u}}_\Pi\,\overline{\mathbf{u}}_\Pi\right) = -\frac{\boldsymbol{\tau}_\Pi}{\rho_{\mathrm{c}}} + r\,\overline{\phi}_\Pi\,h_\Pi\,\mathbf{g}_{\mathrm{s}} - \frac{1}{2}\,\boldsymbol{\nabla}_{\mathrm{s}}^\Gamma \left(\left(1 + r\,\overline{\phi}_\Pi\right)g_{\mathrm{eff}}\,h_\Pi^2\right) + \frac{S_\Pi^{\mathrm{u}}}{\rho_{\mathrm{c}}}. \tag{30}$$

All equations and terms are well known from the dense flow model, except for the additional tracking of grains with Eq. (29).

The unknown flow fields are the flow depth $h_\Pi$, the depth-averaged velocity $\overline{\mathbf{u}}_\Pi$ and the depth-averaged phase fraction or packing density $\overline{\phi}_\Pi$. Assuming $r\,\overline{\phi}_\Pi \approx 0$ in all terms but the gravitational acceleration (buoyancy assumption), leads to the popular model of Parker et al. (1986). Removing the surface tangential gravitational acceleration leads to the momentum conservation equation of Bartelt et al. (2016). The effective gravitational acceleration $g_{\mathrm{eff}}$ is the surface normal gravitational acceleration, corrected for centripetal acceleration due to curved terrain. In terms of surface partial differential equations, it can

be easily expressed as (see appendix A)

$$g_{\mathrm{eff}} \approx \mathbf{n}^\Gamma \cdot \left(\mathbf{g} - \boldsymbol{\nabla}^\Gamma \cdot \left(\overline{\mathbf{u}}_\Pi\,\overline{\mathbf{u}}_\Pi\right)\right). \tag{31}$$

This expression replaces the rather complex calculation of the basal pressure in the dense flow model. It is justified here, as the basal pressure has only a weak influence on the flow dynamics of the suspended flow. Further, this notation turns out to





be convenient later, as various internal processes in the suspension flow are depending on effective gravity. A particle on a
streamline of the flow, will approximately experience a volume force corresponding to this acceleration and processes like the
terminal settling velocity will depend on this adjusted value.

Considerable attention has to be drawn to the volumetric source and sink terms, $S_\Pi^h$ and $S_\Pi^\phi$ and the associated momentum
flux $S_\Pi^u$. These terms are responsible for the varying flow height and the depth-averaged packing density and influence the flow
dynamics substantially.

## 4.1 Friction in the Suspension Flow Model

Similar as in the dense flow model, the term $\boldsymbol{\tau}_\Pi$ represents the depth-integrated divergence of the shear stress tensor. If the
particle fraction in the suspension is low, it can be treated as a simple fluid. Assuming laminar flow, the friction can be calculated
with a constant wall friction coefficient $c_D$ (Parker et al., 1986),

$$\boldsymbol{\tau}_\Pi = \rho_c \, c_D \, |\overline{\mathbf{u}}_\Pi| \, \overline{\mathbf{u}}_\Pi. \tag{32}$$

However, suspension flows are inherently turbulent, reaching Reynolds numbers of up to $10^9$ (Meiburg et al., 2015), as they
need to pick up particles and keep them suspended. Nevertheless we will use the simple laminar model in this work. Under
turbulent conditions, the wall friction coefficient $c_D$ looses its physical meaning and takes the form of an empirical parameter
that might require adaption to flow conditions. Further, it is assumed that all dissipative processes, such as inter-granular
friction (e.g. Boyer et al., 2011), are included in this term. Considering the accuracy and uncertainties of the problems at hand,
this seems to be a reasonable compromise. Alternative approaches are the turbulent friction model of Parker et al. (1986), a
depth-integration of the Einstein viscosity model (e.g. Boyer et al., 2011) or a more complex granular rheology (Boyer et al.,
2011).

## 4.2 Ambient fluid entrainment in the Suspension Flow Model

The volume of the suspension flow will grow due to entrainment of ambient fluid. It is assumed (Parker et al., 1986; Turner,
1986; Ancey, 2004) that ambient fluid entrainment depends solely on the Richardson number, which is given as

$$Ri_\Pi = \frac{r \, g_{\text{eff}} \, \overline{\phi}_\Pi \, h_\Pi}{\overline{\mathbf{u}}_\Pi^2}. \tag{33}$$

In contrast to e.g. Parker et al. (1986), we use the effective surface normal acceleration $g_{\text{eff}}$ instead of the constant gravitational
acceleration $|\mathbf{g}|$ to account for the influence of centripetal forces on particles in the flow. Adjusting the Richardson number
with the centripetal acceleration leads to an increased amount of ambient fluid entrainment if the flow runs over convex terrain
and to a decreased amount if the flow runs over concave terrain.

There are various models for the relation between the Richardson number and the entrainment. Parker et al. (1986) use a
simple, inverse proportional approach,

$$S_{\Lambda \to \Pi}^h = |\overline{\mathbf{u}}_\Pi| \frac{\alpha}{Ri_0 + Ri_\Pi}, \tag{34}$$





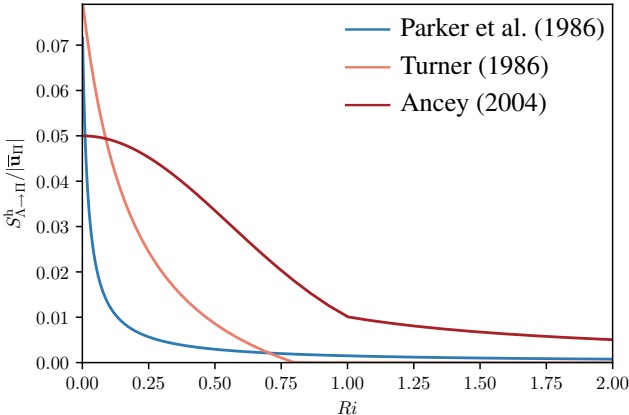

**Figure 4.** Comparison of the air entrainment functions, all depending solely on the Richardson number $Ri$.

with the parameters $\alpha = 0.00153$ and $Ri_0 = 0.204$.

Turner (1986) provides an alternative formulation

$$S_{\Lambda\to\Pi}^{\mathrm{h}} = |\overline{\mathbf{u}}_\Pi| \begin{cases} \dfrac{Ri_0 - Ri_\Pi}{\alpha_1 + \alpha_2 \, Ri_\Pi} & \text{for } Ri_\Pi < Ri_0, \\[2mm] 0 & \text{for } Ri_\Pi \geq Ri_0. \end{cases} \tag{35}$$

with the parameters $Ri_0 = 0.8$, $\alpha_1 = 10$ and $\alpha_2 = 50$. Various different parameters were suggested for this empirical relation, see e.g. Ancey (2004).

     Finally, Ancey (2004) suggested yet another relation in form of an exponential function, here given in the form of Issler et al.
320   (2018)

$$S_{\Lambda\to\Pi}^{\mathrm{h}} = |\overline{\mathbf{u}}_\Pi| \, \alpha_2 \begin{cases} \exp\left(-\alpha_1 \, Ri^2\right) & \text{for } Ri_\Pi < 1, \\[2mm] \exp\left(-\alpha_1\right)/Ri & \text{for } Ri_\Pi \geq 1. \end{cases} \tag{36}$$

The parameter $\alpha_1$ is supposed to be the only free parameter, with a value of $1.6$ following Issler et al. (2018), however, due to different definitions of the entrainment rate an additional parameter $\alpha_2$ is required. In order to be of similar magnitude as the other air entrainment relations, $\alpha_2$ has to be roughly $0.05$. All relations are shown in Fig. 4.

**4.3   Grain entrainment and settlement in the Suspension Flow Model**

Suspension flows are, similar to dense flows, able to erode granular material from the bed. It is, in principle, possible to use the same entrainment relations as in the dense flow model, but specialized entrainment relations have been proposed in literature.





An example for subaquatic turbidity currents, is given by Parker et al. (1986) as

$$
S_{\Sigma\to\Pi}^{\phi} = v_{\mathrm{s}}
\begin{cases}
0.3 & \text{for } Z > Z_{\mathrm{m}}, \\
3 \cdot 10^{-12} \, Z^{10} \left(1 - \frac{Z_{\mathrm{c}}}{Z}\right) & \text{for } Z_{\mathrm{c}} < Z < Z_{\mathrm{m}}, \\
0 & \text{for } Z < Z_{\mathrm{e}},
\end{cases}
\tag{37}
$$

with

$$
Z = Re_{\mathrm{g}} \, \frac{\sqrt{\tau_{\Pi}}}{v_{\mathrm{s}}},
\tag{38}
$$

the settling velocity

$$
v_{\mathrm{s}} = \frac{r \, g_{\mathrm{eff}} \, d_{\Pi}^2}{18 \, \nu_{\mathrm{c}}},
\tag{39}
$$

the particles Reynolds number

$$
Re_{\mathrm{g}} = \frac{\sqrt{r \, g_{\mathrm{eff}} \, d_{\Pi}} \, d_{\Pi}}{\nu_{\mathrm{c}}},
\tag{40}
$$

the viscosity of the ambient fluid $\nu_{\mathrm{c}}$ and two empirical parameters $Z_{\mathrm{m}} = 13.2$ and $Z_{\mathrm{c}}$. The parameter $Z_{\mathrm{c}}$ was reported to be approximately $5$, we found that a value of exactly $0.5$ is required to reproduce the examples of Parker et al. (1986) in the examples shown in section 7.2.

The settling of grains is given by Parker et al. (1986) as

$$
S_{\Pi\to\Sigma}^{\phi} = v_{\mathrm{s}} \, r_0 \, \phi_{\Pi},
\tag{41}
$$

with the settling velocity as given in Eq. (39) and the factor $r_0$ for the bottom value of the grain concentration

$$
r_0 = 1 + 31.5 \left( \sqrt{\frac{\tau_{\Pi}}{\rho_{\mathrm{c}}}} \frac{1}{v_{\mathrm{s}}} \right)^{-1.46}.
\tag{42}
$$

As before, the total flux term follows as the difference between entrainment and deposition,

$$
S_{\Pi}^{\phi} = S_{\Sigma\to\Pi}^{\phi} - S_{\Pi\to\Sigma}^{\phi}.
\tag{43}
$$

The momentum flux into the suspension due to ambient fluid and grain entrainment is zero. The volume occupied by entrained and deposited grains and the respective flux term in the evolution equation of the flow height $h_{\Pi}$ is neglected at this point.

## 5 Two-layer Granular Flow Model

Granular mass flows can show different regimes, especially in terms of the Stokes number. Sampl and Zwinger (2004) and others (Jóhannesson et al., 2009) describe three regimes, the dense flow, transition or re-suspension and powder snow layer, Sovilla et al. (2015) recognize five regions in mixed snow avalanches and Köhler et al. (2018) identified seven regimes. Here we



aim to represent the two limit cases of dense flow and suspension in a single model, similar to Bartelt et al. (2016). It is assumed that these regimes are described in appropriate accuracy either by the Savage and Hutter (1989, 1991) model, Equations (17) to (19), or the Parker et al. (1986) model, Equations (28) to (30). The layers will communicate with mass fluxes $S^\phi$ or $S^h$ and momentum fluxes $\mathbf{S}^u$. In particular, the fluxes of grains are (see also Fig. 1) for the static layer (deposition of the dense flow model is neglected),

$$S^\phi_\Sigma = S^\phi_{\Phi\to\Sigma} - S^\phi_{\Sigma\to\Phi}, \tag{44}$$

for the dense flow layer

$$S^\phi_\Phi = S^\phi_{\Sigma\to\Phi} - S^\phi_{\Phi\to\Sigma} + S^\phi_{\Pi\to\Phi} - S^\phi_{\Phi\to\Pi}, \tag{45}$$

and for the suspension layer

$$S^\phi_\Pi = S^\phi_{\Phi\to\Pi} - S^\phi_{\Pi\to\Phi}. \tag{46}$$

Entrainment from the suspension layer is assumed to be negligible small in comparison to the overall mass fluxes and thus not explicitly accounted for in the simulations. The term $S^\phi_{\Phi\to\Pi}$ describes the upward mass flux from the dense flow to the suspension flow. It is the remaining term to be specified in the following (see section 5.1). The flux in the opposite direction $S^\phi_{\Pi\to\Phi}$ is assumed to be equal to the settling flux of the suspension layer $S^\phi_{\Pi\to\Sigma}$, i.e. the deposition from the suspension is redirected to the dense core and further to the static layer from there, if the deposition model of the dense flow model is active. The corresponding momentum fluxes for the dense flow layer and the suspension layer are

$$\mathbf{S}^u_\Phi = -\mathbf{S}^u_\Pi = \overline{\mathbf{u}}_\Pi\, S^\phi_{\Pi\to\Phi} - \xi_{t\Phi}\, \overline{\mathbf{u}}_\Phi\, S^\phi_{\Phi\to\Pi}, \tag{47}$$

accounting for the momentum that is transferred together with grains between moving layers. The shape factor $\xi_t$ takes into account that the velocity at the top boundary of the avalanche, where particles are tossed into the suspension layer, is higher than the depth-integrated velocity. It is related to the previously shown shape factor and can similarly be calculated on basis of e.g. the Bagnold (1954) velocity profile as $5/3$. The particles that fall from the suspension layer onto the dense flow layer, $S^\phi_{\Pi\to\Phi}$, are assumed to carry the velocity of the suspension layer. The momentum fluxes from and to the static layer are zero due to the respective velocity at the interface.

Further we have to account for the volume of fluid that is pushed into the suspension layer with particles. Assuming that particles enter at a packing density of $\phi_{0\Pi}$, we have to add a source term of the form

$$S^h_{\Phi\to\Pi} = \phi_{0\Pi}\, S^\phi_{\Phi\to\Pi}. \tag{48}$$

The value $\phi_{0\Pi}$ is set to the phase fraction of the dense core in this work. This avoids unreasonably high grain fractions if a suspension flow is initiated by a dense flow avalanche.

In addition to the momentum fluxes, that are related to the mass fluxes, we need to consider the shear stress on the interface. This relation is chosen to be identical to the basal shear stress of the suspension layer, $\tau_\Pi$, however, it is no longer proportional





to the velocity of the suspension layer, but to the relative velocity between the dense flow and the suspension layer,

$$\boldsymbol{\tau}_\Pi = \rho_c \, c_D \, |\overline{\mathbf{u}}_\Pi - \overline{\mathbf{u}}_\Phi| \, (\overline{\mathbf{u}}_\Pi - \overline{\mathbf{u}}_\Phi). \tag{49}$$

In areas where the suspension layer detaches from the dense flow, the dense flow velocity is assumed to be zero and the model collapses to the friction model of the ordinary suspension model. An equal but opposite stress term to $\boldsymbol{\tau}_\Pi$ should be applied to the dense core to account for the friction of the top surface of the dense core. However, it is assumed that this stress is already included in the empirical formulation and parametrisation of $\boldsymbol{\tau}_\Phi$, because the top surface friction is also present in pure dense snow avalanches with a stationary or moving air layer above it. The ambient fluid entrainment of the suspension layer stays unchanged.

The mass flux $S^\phi_{\Phi \to \Pi}$ feeds the suspension layer from the dense core and the associated momentum flux, in combination with the shape factor propels the suspension flow forwards. This is assumed to be the mayor genesis mechanism for the suspension cloud, similar to Bartelt et al. (2016).

## 5.1 Cross-layer coupling

All fluxes of the two layer model are described relatively well in literature (see sections above), except for the mass flux from the dense flow layer to the suspension layer, $S^\phi_{\Phi \to \Pi}$, for which only few suggestions can be found (Sampl and Zwinger, 2004; Bartelt et al., 2016). Existing relations do conceptually not fit into the presented framework, either due to missing granular mechanics (Sampl and Zwinger, 2004) or due to their dependence on a specific dense flow model (Bartelt et al., 2016). For the purpose of introducing this framework we choose a simple relation, based on local flow fields of the dense flow.

We assume that the dense flow is composed of small and large particles with diameter $d_\Pi$ and $d_\Phi$, respectively. Uptake of particles into the suspension layer requires small particles to be made available by the dense layer that mostly consists of large particles (Bartelt et al., 2016), and the capability of the suspension layer to keep them suspended. The latter is already implemented into the model in form of the settling model of the suspension flow $S^\phi_{\Pi \to \Phi}$. This term is depending on the Reynolds particle number $Re_g$ which is similar to the Stokes number and a good indicator for the flow regime.

The first step, making small particles available to the suspension, is assumed to be triggered by a fluidized flow that is expanding in volume, sucking in air and increasing the distance between particles. There are various hints on how this expression should look like. At first it is useful to look at dimensionless properties in the dense flow. Beside the non-dimensional volumetric mass flux $S^\phi_{\Phi \to \Pi}/|\overline{\mathbf{u}}_\Phi|$, these are (Forterre and Pouliquen, 2008; Rauter, 2021) the friction coefficient $\mu_\Phi = |\boldsymbol{\tau}_\Phi|/p_\Phi$, the packing density $\phi_\Phi$ and the inertial number

$$I_\Phi = \frac{d_\Phi \, \dot{\gamma}_\Phi}{\sqrt{p_\Phi/\rho_g}}, \tag{50}$$

with the shear rate at the bottom of the dense flow (Bagnold, 1954)

$$\dot{\gamma}_\Phi = \frac{4}{3} \frac{|\overline{\mathbf{u}}_\Phi|}{h_\Phi}. \tag{51}$$



It is well established that $\mu_\Phi$ and $\phi_\Phi$ can be expressed as a function of only the inertial number $I_\Phi$ and it is reasonable to assume that fluidisation can be described the same way. This is further emphasized by the linear relationship between the packing density and the inertial number in the dense flow regime (Forterre and Pouliquen, 2008). Finally, Rauter et al. (2016) found a specific relation between the shear rate $\dot{\gamma}_\Phi$ and the pressure $p_\Phi$ in a granular kinetic theory model (Vescovi et al., 2013) at the point where fluidisation suddenly occurs,

$$\frac{\dot{\gamma}_\Phi}{p_\Phi^{0.37}} = \text{const.} \tag{52}$$

Comparing this relation to the expression for the inertial number, one can observe a striking resemblance, solely the exponent of the pressure is slightly lower in the relation of Rauter et al. (2016). This strongly indicates that the mass flux from the dense flow to the suspension can be expressed as a function of the inertial number only, starting at a minimum value $I_0$ and growing with a specified rate $s_\mathrm{f}$ from thereon

$$\frac{S_{\Phi\to\Pi}^\phi}{|\overline{\mathbf{u}}_\Phi|}\left(I_\Phi\right) = \max\left(I_\Phi - I_0, 0\right) s_\mathrm{f}. \tag{53}$$

The results of Rauter et al. (2016) suggest that the value of $I_0$ is close to $0.5$, as at this point explosive fluidisation starts to occur. The factor $s_\mathrm{f}$ is expected to be small, as the vertical velocity has to be substantially smaller than the flow velocity. This parameter can be optimized to yield the correct relation between dense flow and powder cloud.

In this model, small particles will be made available to the suspension when the dense flow velocity is high or when the pressure is low, e.g. when an avalanche is running over a bump. If the small particles are sufficiently small, the suspension will be able to keep the particles suspended and a powder cloud will form. Otherwise, the particles will fall back to be reintegrated by the dense core, expressed by the deposition mass flux of the suspension layer, which is stronger for larger particles. The parameters for the suggested model are the small and large particle diameters $d_\Pi$ and $d_\Phi$, the minimum value of $I$ at which fluidisation occurs $I_0$, the particle density $\rho_\mathrm{g}$ and the factor $s_f$. All parameter except for the latter are already used in the model or known otherwise.

Relation (53) finally completes the model and closes the system that will be solved numerically in the following. The model could be improved by tracking and limiting the availability of small particles or by making this property temperature-dependent.

## 6 Pre- and Postprocessing

The pre- and postprocessing of simulations with the presented models follows the workflow depicted by Rauter et al. (2018). The capabilities of respective tools have been improved and fully implemented in C++ to allow a seamless integration into OpenFOAM and computational clusters that do not support Python and some of the previously used libraries. Most improvements are based on a native implementation of two common geographic information system (GIS) data types, ESRI® shape files and ESRI® grid files. The native implementation allows all solvers and utilities of the OpenFOAM avalanche module to directly read and write from or to the respective files. This enables many previously difficult tasks that are presented in the following. Generally, all tools are steered with text files that follow the usual OpenFOAM syntax, called dictionary (see Fig. 5).





**Figure 5.** Pipeline of the OpenFOAM avalanche module. The pipeline has been simplified substantially since the work of Rauter et al. (2018). Most notably, all components are fully implemented in C++ and included into the module. The pipeline includes the complete workflow, starting from GIS data and returning all results to GIS data. The user can modify parameters in the respective dictionaries and geometry of the simulation domain and the initial conditions in the geographic input data. The solver can be replaced with any of the three models.

This toolchain-based workflow follows the concept of OpenFOAM, which has proven to enable reproducibility, reusability but also rapid development.





## 6.1 Mesh Generation from Terrain Data

The mesh generation follows the principles from Rauter et al. (2018). In a first step a triangulation of the terrain and a boundary of the surrounding volume is generated. A new tool for this task, called `gridToSTL` was written entirely in C++ and without any external dependencies. The tool requires input in form of a polygon that defines the simulation domain and the terrain data in form of a raster file. Other than in the previous version, the polygon can be any kind of closed and non-intersecting polygon with an arbitrary number of edges, either convex or concave. This enables flexibility on the simulation domain, which turned
out to be especially useful to cover long and windy submarine canyons.

The finite volume mesh is generated from the triangulated surface with an arbitrary mesh generator. This toolchain can not only be applied to the depth-integrated models presented here but was also used for the full three-dimensional model presented by Rauter et al. (2022). In this study we used the mesh generator `pMesh`, while Rauter et al. (2022) used `cartesianMesh`, both of the cfMesh toolbox (Juretić, 2015). The finite area mesh is then generated on a dedicated surface of the finite volume
mesh using the tool `makeFaMesh`, part of the OpenFOAM finite area module.

## 6.2 Mapping Initial Conditions

Initial conditions can be set with the tool `releaseAreaMapping`. In addition to the functionality of previous versions, this tool is now able to read shape files and grid files and map them directly onto finite area fields to be used by any solver. All input for the tool is read from a dictionary, where further references to shape and grid files can be listed. This tool enables efficient
adaptions to new scenarios.

## 6.3 Simulation Run

Once the mesh and the initial conditions are defined, the solver of choice can be run. Currently there are three solvers available in the avalanche module, the dense flow solver `faSavageHutterFoam`, the suspension flow solver `faParkerFukushimaFoam` and the mixed flow solver `faTwoLayerAvalancheFoam` (Fig. 5 shows `faTwoLayerAvalancheFoam` only, but it can
be replaced with any other model). Physical parameters are read from the file `transportProperties`, general simulation settings are read from the `controlDict` and numerical algorithms and parameters from the files `faSolution` and `faSchemes`. To run the solver in parallel, the tool `decomposePar` has to be run before the solver and the tool `reconstructPar` has to be run after the solver. In the common OpenFOAM manner, all steps for a simulation are listed in a script file named `Allrun`, which can executed by the user to automatically execute the here proposed pipeline. Another
script, named `Allclean` can be run to clean up the simulation directory.

## 6.4 Postprocessing and Data Export

The OpenFOAM architecture allows to execute customized code, called function objects in every simulation step. Various function objects are made available in the avalanche module. Most importantly, this includes function objects to export simulation results as either shape or raster files. The export as shape files can be done cell-wise (one polygon for each computational





cell) or the numerical data can be recombined to generate iso lines that are written into the shape file. Function objects can
be loaded by placing the respective entry in the control dictionary. As of version v2312, all solvers are able to run in a post-
processing mode, in which old results are read from hard disc and the function objects are executed. This allows to execute
function objects in a post-processing workflow without rerunning the whole simulation.

## 7   Results and Discussion

### 7.1   Dense Flow Model


The dense flow model was applied to various cases in multiple studies. The interested reader is referred to Rauter and Tuković
(2018) for lab scale simulations, Rauter et al. (2018) and Huber et al. (2018) for large scale snow avalanche simulations, Rauter
and Köhler (2020) for simulations with the deposition model and to Shimizu (2022) for an application to pyroclastic flows.

### 7.2   Suspension Flow Model

Parker et al. (1986) simulate steady suspension flows on constantly inclined one-dimensional slopes with the model presented in
section 4. Four cases with uniform model parameters but different boundary conditions give a good overview over the behaviour
of the model and a verification (as defined by Roache, 1997, as solving the equations right) of the presented implementation.
The four simulations are conducted on one-dimensional slopes with a gradient of $5\%$, the gravitational acceleration follows as
$\mathbf{g} = (0.49, 0, -9.81)^T\,\mathrm{m\,s^{-2}}$ (chosen to match the setup by Parker et al., 1986). The parameters suggest that the suspensions
are composed of sediment in water on a scale of a small turbidity current.

Material parameters for this setup are given in Tab. 1. The left boundary condition (at $x = 0$) prescribes the inflow in terms of
the height $h_\Pi$, velocity $\overline{\mathbf{u}}_\Pi$ and grain flux $\overline{\psi}_\Pi = h_\Pi\,\overline{\phi}_\Pi\,\overline{\mathbf{u}}_\Pi$, in particular as shown in Tab. 2. All parameters are given normalized
to reference values $H = 2\,\mathrm{m}$, $U = 0.874\,\mathrm{m\,s^{-1}}$ and $\Psi = 0.00828\,\mathrm{m^2\,s^{-1}}$.

The right boundary condition is modelled as zero gradient for all fields, mimicking an outlet boundary condition. For a basic
verification of the novel implementation of the suspension model, the respective simulations are repeated and compared to the
original results. We will evaluate the buoyancy assumption of Parker et al. (1986), as well as the formulation with the correct
density given in here. The simulations are conducted in an unsteady manner until the flow reaches a steady state, comparable
to the results reported by Parker et al. (1986). Figure 6 shows results for the four cases.

The first case, starting with a high velocity but low particle fraction increases its particle fraction quickly, as the high velocity
is sufficient to erode and pick up sediment. The second case starts with a very high phase fraction, leading to a sudden ignition
of the flow at $x/H = 60$. The height of the suspension stays low and even decreases, showing that a high phase fraction can
keep the suspension concentrated at the bottom. The third and fourth case start with a low velocity and low particle phase
fraction, respectively, and the suspension fades out quickly. The height of the flow is increasing in both cases where the flow
is fading out, indicating that the momentum of the flow is diffused over larger volumes of fluid. This is consistent with the
expected scaling of fluid entrainment with the Richardson number.



**Table 1.** Parameters for the small scale simulations of Parker et al. (1986).

| sub model | parameter | description | value |
|---|---|---|---|
| flow model: | | **Parker–Fukushima** | |
| | $\rho_\mathrm{s}$ | density of solid phase (particles) | $2650\,\mathrm{kg\,m^{-3}}$ |
| | $\rho_\mathrm{c}$ | density of fluid phase (water) | $1000\,\mathrm{kg\,m^{-3}}$ |
| | $r$ | density ratio, follows as | 1.65 |
| | $\nu_\mathrm{c}$ | viscosity of fluid phase (water) | $10^{-6}\,\mathrm{m^2\,s^{-1}}$ |
| | $d_\Pi$ | particle diameter | $10^{-4}\,\mathrm{m}$ |
| basal friction: | | **laminar flow** | |
| | $c_\mathrm{D}$ | drag coefficient | 0.004 |
| particle entrainment: | | **Parker–Fukushima** | |
| | $Z_\mathrm{c}$ | empirical parameter | 0.5 |
| | $Z_\mathrm{m}$ | empirical parameter | 13.2 |
| ambient fluid entrainment: | | **Parker–Fukushima** | |
| | $Ri_0$ | reference Richardson number | 0.0204 |
| | $\alpha$ | reference ambient fluid entrainment | 0.00153 |
| deposition: | | **Parker–Fukushima** | (no parameters) |

**Table 2.** Inlet boundary conditions for the small scale simulations of Parker et al. (1986), simulating four scenarios of igniting or fading turbidity currents.

| case | $h_\Pi/H$ | $\overline{\mathbf{u}}_\Pi/U$ | $\overline{\psi}_\Pi/\Psi$ |
|---|---|---|---|
| (a) | 1.0 | 1.3 | 0.2 |
| (b) | 1.0 | 0.9 | 1.7 |
| (c) | 1.0 | 0.7 | 1.2 |
| (d) | 1.0 | 1.0 | 0.2 |

It can be seen that results of Parker et al. (1986) are reproduced with only small derivations. The OpenFOAM solver yields sharper edges than the the implementation of Parker et al. (1986), especially visible in Fig. 6b. This small difference is most likely attributed to the numerical solution method or the numerical resolution. The correction of the time derivative and advection term with $(1 + r\,\overline{\phi}_\Pi)$ has only a minimal influence on the model results. This is reasonable, considering the low value for the buoyant density ration $r = 1.65$ in these cases. These simulations provide a strong indicator that the model of Parker et al. (1986) was implemented correctly, however, this can not be seen as a validation (Roache, 1997) of the model.



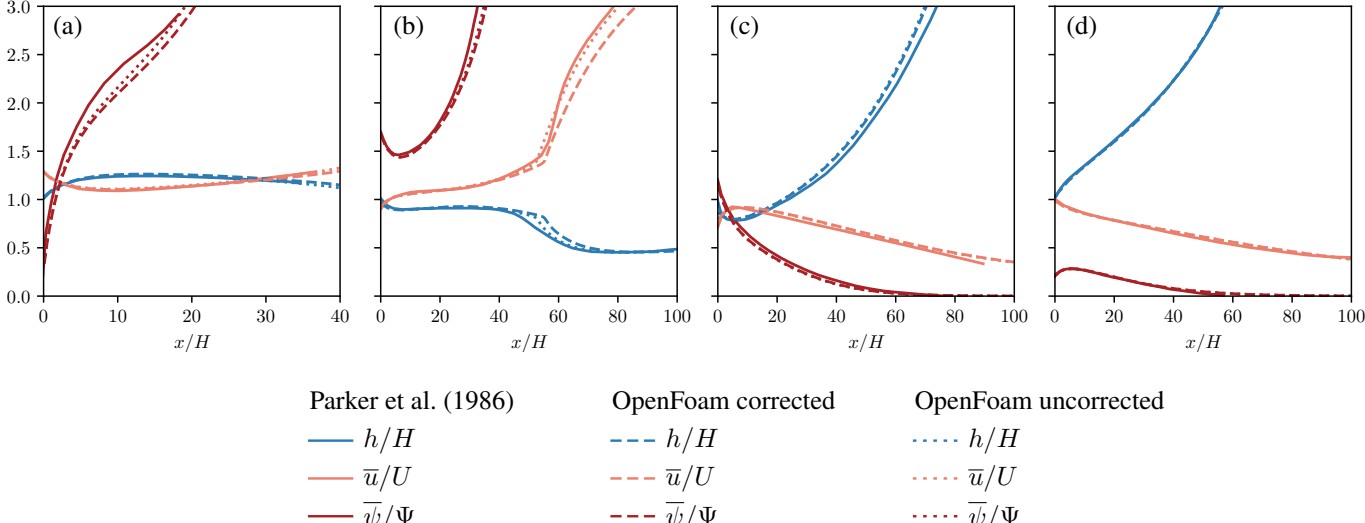

**Figure 6.** Numerical simulation of the four test cases presented by Parker et al. (1986) with OpenFOAM, with and without the buoyancy assumption (corrected and uncorrected, respectively). The results of Parker et al. (1986) are reproduced with good accuracy. The buoyancy assumption fits well to the conditions of these numerical experiments.

## 7.3 Two Layer Model

### 7.3.1 Synthetic tests and sensitivity study

In order to better understand the two layer dense flow - suspension flow model, we will conduct tests on synthetic topographies.
The topography is based on a parabola with a length $L = 4\,000\,\mathrm{m}$ and a height $H = 2\,000\,\mathrm{m}$, with an additional flat runout area of $2\,000\,\mathrm{m}$. The slope has a width of $2\,000\,\mathrm{m}$, leading to a simulated region of $x = [-4\,000, 2\,000]\,\mathrm{m}$ and $y = [-1\,000, 1\,000]\,\mathrm{m}$. In addition, the influence of topographic structures will be investigated, as terrain features often initialize the formation of suspension flows, e.g. powder snow avalanches. A bump in the surface is created by superposing the parabola with a secans hyperbolicus $\mathrm{sech}\,(x) = 2/\left(\exp(x) + \exp(-x)\right)$ at $X_\mathrm{p} = -2\,700\,\mathrm{m}$ with height $H_\mathrm{p} = 150\,\mathrm{m}$ and length $L_\mathrm{p} = 200\,\mathrm{m}$,

$$
\quad z = H_\mathrm{p}\,\mathrm{sech}\left(\frac{x - X_\mathrm{p}}{L_\mathrm{p}}\right) + 
\begin{cases}
H\left(\dfrac{x}{L}\right)^2 & \text{for } x < 0, \\
0 & \text{otherwise,}
\end{cases}
\tag{54}
$$

inspired by the experiments of Viroulet et al. (2017). All boundaries are implemented as Neumann (zero gradient) boundary conditions.

The release area (initial condition) of the slide was formed by a square between $x = [-3\,900, -3\,500]\,\mathrm{m}$ and $y = [-500, 500]\,\mathrm{m}$ and an initial dense flow height of $h_\Phi = 5\,\mathrm{m}$ within that square. All other flow fields are set to zero. The parameters, roughly
525 corresponding to snow avalanches are given in Tab. 3, if not mentioned otherwise. The value for the coupling factor $s_\mathrm{f}$ is varied



and the sensitivity of the model to this parameter is investigated. Entrainment and deposition from and to the static layer are not included in this section for simplicity. The simulations were run for $90\,\mathrm{s}$.

Beside the flow thickness, velocity and phase fraction, we can analyse the dynamic pressure, which is an important indicator for the destructive potential of the flow. It is defined as

$$p_{\mathrm{d}\,\Phi} = \rho_\Phi \left| \overline{\mathbf{u}}_\Phi \right|^2 \tag{55}$$

for the dense flow and as

$$p_{\mathrm{d}\,\Pi} = \left( \rho_\mathrm{g}\,\phi_\Pi + \rho_\mathrm{c}\,(1 - \phi_\Pi) \right) \left| \overline{\mathbf{u}}_\Pi \right|^2 \tag{56}$$

for the powder cloud (e.g. Jóhannesson et al., 2009). In particular we evaluate the dynamic peak pressure, which the maximum of the dynamic pressure at a fixed point over time. Important limits that are used in the definition of hazard zones, e.g. in Austria, are $1\,\mathrm{kPa}$ (yellow zone) and $10\,\mathrm{kPa}$ (red zone) (Jóhannesson et al., 2009). Notably, the shape factor should be applied to the dynamic pressure for consistency, increasing all simulated pressures by $25\%$. However, this is neglected in order to be consistent with previous works and the definition of hazard zones.

Results for a simple parabola (without surface bump) are shown in Fig. 7 for three values of $s_\mathrm{f}$ ($10^{-5}$, $10^{-4}$, $10^{-3}$). This set of simulations allows some valuable conclusions on the model and in particular the coupling model. All simulations start with a dense flow that eventually feeds the powder cloud. The feed of the powder cloud varies strongly due to the variation of the respective parameter $s_\mathrm{f}$.

For a low value of $s_\mathrm{f}$ the dense flow is not able to generate a strong powder cloud with a considerable phase fraction and thus density. A suspension flow develops eventually, however, it consists almost entirely of air, without any ice particles. Basically, this can be seen as a layer of air that is dragged along by the dense flow. The velocity, dynamic pressure and runout distance of this layer are respectively low. As shown before, the flow height of the suspension layer grows strongly for fading flows, indicating a strong diffusion of momentum.

Increasing the value for $s_\mathrm{f}$ up to $10^{-4}$ leads to higher phase fractions up to $0.004$, roughly corresponding to a density of $4\,\mathrm{kg\,m^{-3}}$. Further increasing the value to $10^{-3}$ leads to phase fractions of up to $0.02$ and densities of $20\,\mathrm{kg\,m^{-3}}$, however only for short periods. Notably, these are depth-averaged phase fractions and densities and the respective values close to the surface might be considerably higher. The respective dynamic pressure of the powder cloud is still low and only the simulation with the highest coupling factor $s_\mathrm{f}$ is able to generate a red zone that extends beyond the red zone of the dense flow. These results seem reasonable, considering the relatively low average slope gradient of $50\%$ and the absence of any topographic features that might enhance the feed of the powder cloud. More powerful powder snow avalanches can be expected on steeper slopes and on slopes with high topography variations, e.g. steep cliffs or rough terrain. Further simulations (not shown here) revealed that the powder cloud increases substantially with higher slope gradients.

Results for the slope with a bump are shown in Fig. 8. The model shows a high sensitivity to the terrain and this case represents natural slopes with varying gradients better. All simulations create a considerable powder clouds with high phase fractions. The highest phase fraction is reached shortly after the top of the bump where the negative centrifugal forces are





**Table 3.** Parameters for the two-layer model for synthetic cases on parabolas and for the Wolfsgruben and Eiskar avalanches.

| sub model | parameter | description | parabola | Wolfsgrube | Eiskar |
|---|---|---|---|---|---|
| | $\rho_\mathrm{s}$ | Density of solid phase (snow/ice) | $800\,\mathrm{kg\,m^{-3}}$ | $800\,\mathrm{kg\,m^{-3}}$ | $800\,\mathrm{kg\,m^{-3}}$ |
| | $\rho_\mathrm{c}$ | Density of fluid phase (air) | $1.25\,\mathrm{kg\,m^{-3}}$ | $1.25\,\mathrm{kg\,m^{-3}}$ | $1.25\,\mathrm{kg\,m^{-3}}$ |
| | $\nu_\mathrm{c}$ | Viscosity of fluid phase (air) | $1.5\cdot10^{-5}\mathrm{m^2\,s^{-1}}$ | $1.5\cdot10^{-5}\mathrm{m^2\,s^{-1}}$ | $1.5\cdot10^{-5}\mathrm{m^2\,s^{-1}}$ |
| dense flow: | | **Savage–Hutter model** | | | |
| | $\phi_\Phi$ | Packing density in the dense flow | 0.25 | 0.25 | 0.25 |
| | $d_\Phi$ | large particle diameter | $10^{-2}\,\mathrm{m}$ | $10^{-2}\,\mathrm{m}$ | $10^{-2}\,\mathrm{m}$ |
| | $\xi_\Phi$ | Shape factor | 1.25 | 1.25 | 1.25 |
| | $\xi_{\mathrm{t}\Phi}$ | shape factor for velocity at top | 1.67 | 1.67 | 1.67 |
| dense flow friction: | | **Simplified Kinetic Theory** | | | |
| | $\mu$ | Dry friction coefficient | 0.25 | 0.26 | 0.20 |
| | $\chi$ | dynamic friction coefficient | $10^4\mathrm{m^{-1}\,s^{-2}}$ | $8\,700\mathrm{m^{-1}\,s^{-2}}$ | $10^4\mathrm{m^{-1}\,s^{-2}}$ |
| dense flow entrainment: | | **Erosion energy** | | | |
| | $e_\mathrm{b}$ | Erosion energy | | $10^3\,\mathrm{m^2\,s^{-2}}$ | $10^3\,\mathrm{m^2\,s^{-2}}$ |
| powder cloud: | | **Parker–Fukushima model** | | | |
| | $d_\Pi$ | small particle diameter | $10^{-4}\,\mathrm{m}$ | $10^{-4}\,\mathrm{m}$ | $10^{-4}\,\mathrm{m}$ |
| | $\xi_\Pi$ | shape factor | 1.25 | 1.25 | 1.25 |
| powder cloud friction: | | **Laminar flow** | | | |
| | $c_\mathrm{D}$ | drag coefficient | 0.5 | 0.5 | 0.1 |
| ambient fluid entrainment: | | **Parker–Fukushima** | | | |
| | $Ri_0$ | reference Richardson number | 0.0204 | 0.0204 | 0.0204 |
| | $\alpha$ | reference air entrainment factor | 0.00153 | 0.00153 | 0.00153 |
| powder cloud deposition: | | **Parker–Fukushima** | (no parameters) | | |
| coupling: | | **Inertial number scaling** | | | |
| | $I_0$ | reference inertial number | 0.5 | 0.5 | 0.5 |
| | $s_\mathrm{f}$ | reference suspension feed factor | $10^{-5}$ | $10^{-5}$ | $10^{-4}$ |

strongest and the basal pressure the lowest. The phase fraction reaches up to $0.05$, roughly corresponding to a density of
$50\,\mathrm{kg\,m^{-3}}$. A shock is formed at the bump in the suspension layer due to the high gradient in the phase fraction, leading to a considerable pressure gradient that decelerates the flow. In all simulations the dynamic powder cloud pressure exceeds $10\,\mathrm{kPa}$ and the respective high pressure zone extends beyond the dense flow runout. The $1\,\mathrm{kPa}$ zone of the powder cloud reaches considerable runouts beyond the dense flow.

The results on synthetic terrain show a reasonable behaviour of the model, both in terms of parametrisation and response
to the terrain. The effect of the terrain is well visible and corresponds to the assumptions from which the model was derived. The sensitivity of the model to the parameter $s_\mathrm{f}$ is well pronounced and this factor can be utilized to fit the model to real world observations. A value between $10^{-4}$ and $10^{-5}$ seems reasonable for the parameter of the coupling model.



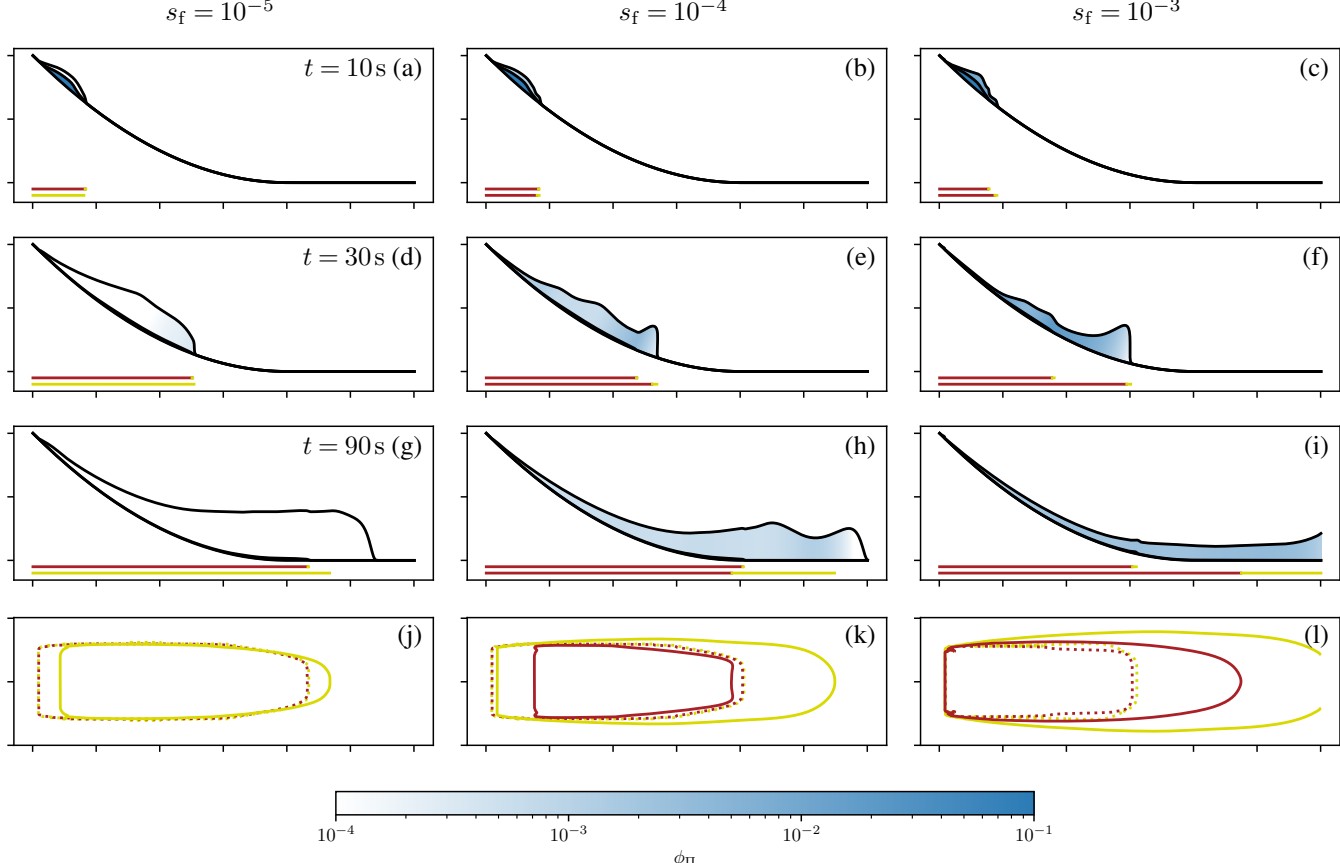

**Figure 7.** Numerical simulations of snow avalanches on a parabolic slope with the two layer model. The parameter $s_{\mathrm{f}}$ was varied between $10^{-5}$ (a,d,g,j), $10^{-4}$ (b,e,h,k) and $10^{-3}$ (c,f,i,l). Panels (a)-(i) show the cross section in the middle of the slide. The slope is shown as the lower black line. The flow thickness $h_{\Phi}$ is shown as offset from the surface magnified by a factor of 20, the flow thickness $h_{\Pi}$ is shown above the dense flow magnified by a factor of 10. The powder cloud is coloured according to the phase fraction $\phi_{\Pi}$. The red and yellow lines below the slope mark the regions of high dynamic peak pressure $p_{\mathrm{d}} > 10\,\mathrm{kPa}$ and intermediate dynamic pressure $p_{\mathrm{d}} > 1\,\mathrm{kPa}$ for the dense flow (top) and the powder cloud (bottom) respectively. Panels (j)-(l) show the regions of high and intermediate dynamic peak pressure (dashed: dense flow, continuous: powder cloud) from the top. One tick on the axis equals $1\,000\,\mathrm{m}$.

Finally, we will use synthetic cases to showcase the sensitivity of the model to the air entrainment. Figure 9 shows the simulation on the synthetic terrain with the three presented air entrainment models. The differences are small but noticeable.
In particular, the entrainment is stronger with the model of Ancey (2004), however, which is just a question of parametrisation. More importantly, the model of Turner (1986) shows a more pronounced flow head. The Richardson number is low in the head and the relation of Turner (1986) predicts the strongest entrainment at low Richardson numbers, see Fig. 4. Generally, all relations appear reasonable and well in line with each other. We will continue with the entrainment model of Parker et al.





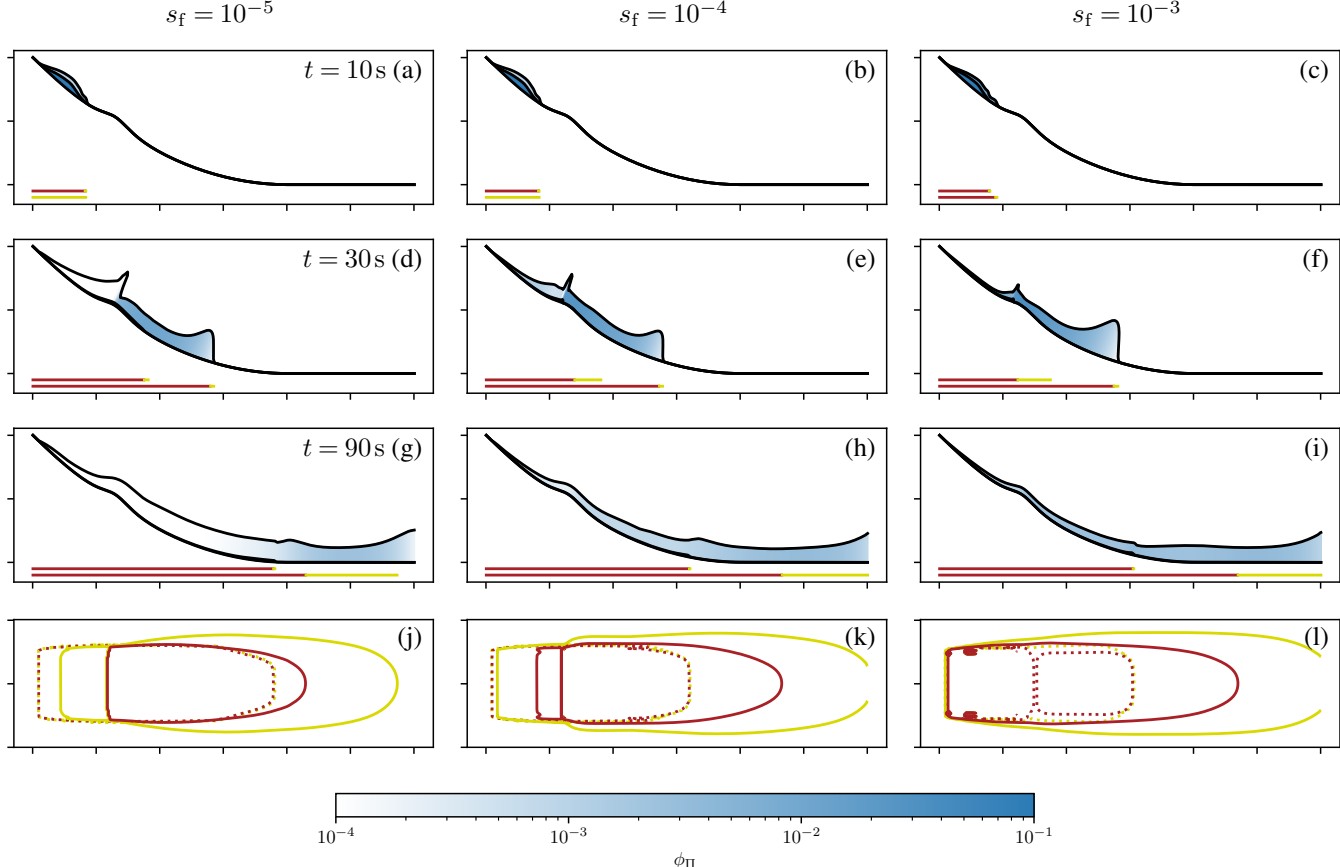

**Figure 8.** Numerical simulations of snow avalanches on a parabolic slope wit a bump with the two layer model. Same as Fig. 7 but with a bump with height $150\,\mathrm{m}$ and length $200\,\mathrm{m}$ at $x = -2700\,\mathrm{m}$.

(1986) from here on. Considering an optimisation of air entrainment parameters to real events, it might be useful to apply the
model of Ancey (2004) instead, as it provides the clearest parametrisation.

### 7.3.2 Real case example: The 1988 Wolfsgruben Avalanche

The 1988 Wolfsgruben Avalanche represents an important event in Austria, as it was the trigger for many developments and used repeatedly as a benchmark. The event, or at least its dense core, was featured by Fischer et al. (2015) and Rauter et al. (2018). Here we revisit the event with the new two layer model and include the powder cloud into the analysis. The avalanche
is characterized by a channelised, steep slope with an angle of $30°$ that transitions quickly into the flat valley floor and the opposite slope.

The preprocessing and simulation setup follows Rauter et al. (2018) but with the novel tool-chain and an extended simulation domain to cover the full runout of the powder cloud. The initial release area of the avalanche and the erodible snow covers are





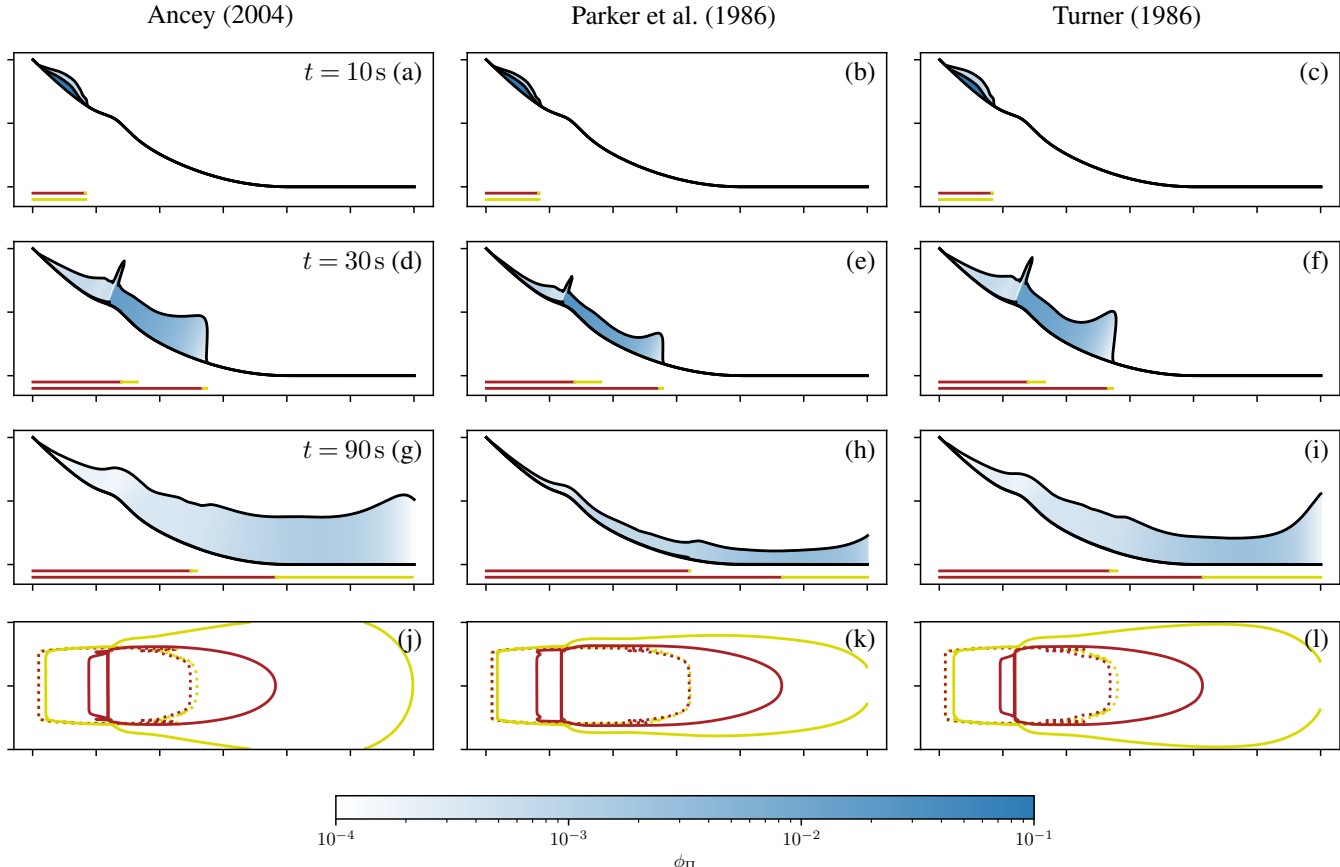

**Figure 9.** Numerical simulations of snow avalanches on a parabolic slope with a bump. Same as Fig. 8 but with a variation of the air entrainment model and a fixed parameter $s_f$.

the same, following the linear approach

$$h_\Sigma|_{t=0}(z) = \left( h_\Sigma(z_0) + \frac{\partial h_\Sigma}{\partial z} (z - z_0) \right) \cos(\theta), \tag{57}$$

where $z$ is the surface elevation and $z_0$ the elevation of a reference point with the base value $h_\Sigma(z_0)$. The growth rate $\frac{\partial h_\Sigma}{\partial z}$ defines the evolution from that point. $\theta$ is the angle between the gravitational acceleration and the surface-normal vector. For the 1988 Wolfsgruben Avalanche we use the snow cover parameters $h_\Sigma(z_0) = 1.61\,\mathrm{m}$, $z_0 = 1289\,\mathrm{m}$, $\frac{\partial h_\Sigma}{\partial z} = 8 \cdot 10^{-4}$.

The model parameters are shown in Tab. 3. The dense flow parameters have been optimized in a previous study (Fischer et al., 2015) and although we use a slightly different friction model, the parameters fit the case well. The suspension parameters are deduced from literature where possible (density, grain diameter). The coupling parameter $s_f = 10^{-5}$ was found after running some simulations, starting from the values derived from the simulations on synthetic cases. A higher value lead to an unrealistically short dense flow runout, a lower value to a severe underestimation of the suspension impact pressure. The





friction coefficient $c_D$ was chosen sufficiently large for the powder cloud to not completely decouple from the dense core.
Apart from this effect, the simulation is rather insensitive to the friction coefficient $c_D$.

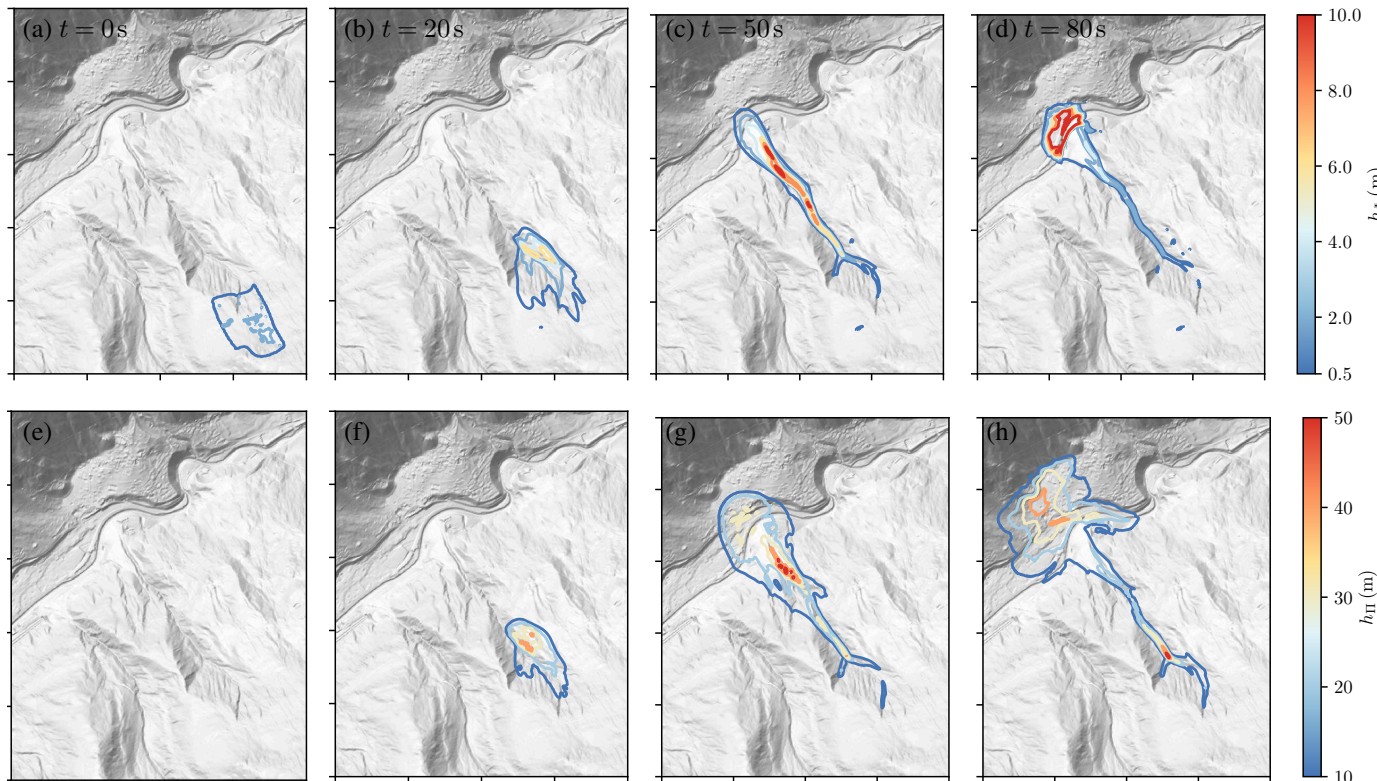

**Figure 10.** Numerical simulation of the Wolfsgruben avalanche with the two layer model. The first row (a-d) shows the height of the dense flow layer, the second row (e-h) shows the height of the powder cloud layer. Each tick on the x- and y-axes corresponds to $500\,\mathrm{m}$.

Four time steps of the simulation are shown in Fig. 10, displaying isolines of the dense flow height $h_\Phi$ (a-d) and the suspension flow height $h_\Pi$ (e-h). The avalanche starts as a dense flow and rapidly accelerates due to the steep release area (Fig. 10a). Shortly after the release a strong suspension layer is formed that further accelerates beyond the velocity of the dense flow layer (Fig. 10b,f). After roughly $40-50\,\mathrm{s}$ the avalanche reaches the bottom of the valley (Fig. 10c,g). The powder cloud
outruns the dense flow and hits the valley floor first. The dense flow is stopped quickly due to the high granular friction while the powder cloud keeps running up on the opposite slope for approximately $50\,\mathrm{m}$ of elevation. Both flows experience a shock that increases the flow height in the valley floor drastically. The deposition, i.e. the dense flow height in the last time step, reaches up to $15\,\mathrm{m}$, however, which does not account for the difference between flow ($\approx 200\,\mathrm{kg\,m^{-3}}$) and deposition density ($\approx 600\,\mathrm{kg\,m^{-3}}$).
Results for the dense flow can be validated by a comparison with the deposition (see Fig. 11b) and they are similar to previous studies with the same model and the model SamosAT (Fischer et al., 2015; Rauter et al., 2018). Results for the powder cloud





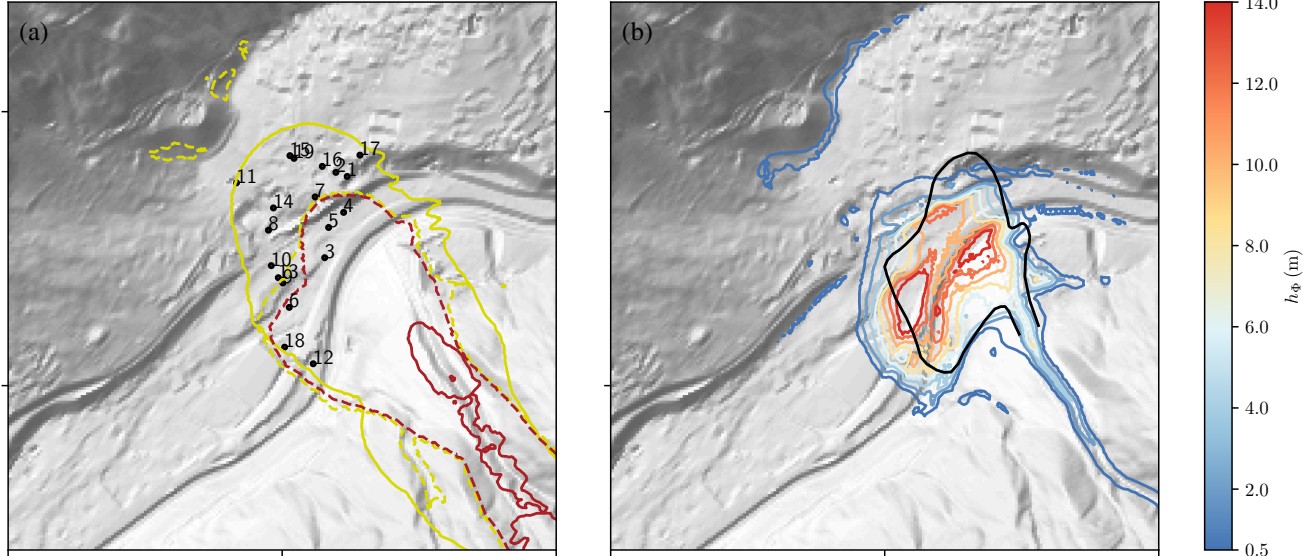

**Figure 11.** (a) The dynamic peak pressure of the suspension layer (solid lines) and the dense flow layer (dashed line). The yellow line marks the $1\,\mathrm{kPa}$ isoline and the red line the $10\,\mathrm{kPa}$ isoline. (b) The deposition of the avalanche (at $t = 180\,\mathrm{s}$).

are more difficult to validate. Traces of the powder cloud that can be identified in the field are limited and not straight forward to interpret, as no clear deposition pattern emerges from suspended flows. Further, the respective deposition can hardly be related to the impact pressure and thus the destructive potential of the flow. Therefore, we compare the simulated dynamic pressure with observed building damages from the respective avalanche (see Fig. 11a). This includes not only the suspension layer but also the dense flow. An evaluation of the dynamic peak pressure and the deposition height at damaged objects is shown in Tab. 4.

The dense flow does not reach the two destroyed buildings (Point 1 and 2 in Fig. 11a and Tab. 4) and stops about $20\,\mathrm{m}$ short. Points 12 and 18 were only slightly damaged by the suspension flow in reality but severely hit by the dense flow in the simulation, showing that the simulation tends too strongly to the left side (viewed in flow direction). The deposition height that was recorded at selected points (Tab. 4) is matched well, assuming a compaction of the avalanche by a factor of 3 after deposition.

The suspension layer shows a very limited zone of high dynamic pressure ($> 10\,\mathrm{kPa}$) but an extended zone of intermittent dynamic pressure ($1 - 10\,\mathrm{kPa}$). The model predicts dynamic pressures of $1 - 4\,\mathrm{kPa}$ where balconies and roofs have been damaged and $1 - 3\,\mathrm{kPa}$ where windows have been destroyed. This corresponds well with engineering estimations of resistance capabilities of the respective parts: Windows are assumed to break at $2 - 4\,\mathrm{kPa}$, doors, walls and roofs at $3 - 6\,\mathrm{kPa}$ (Sovilla et al., 2015). The dynamic pressure of the suspension layer at the destroyed buildings (Point 1 and 2 in Fig. 11) is not sufficient to destroy the respective brick structures ($25 - 45\,\mathrm{kPa}$). These high values strongly indicate that the dense flow or an intermittent regime must be responsible for these high impact pressures (see pictures in Fischer et al., 2015). Therefore, we conclude that



**Table 4.** Simulated dynamic peak pressure at the location where damage was observed.

| Number | $h_\Phi$ | $p_{d\Phi}$ | $p_{d\Pi}$ | Observed damage |
|---|---|---|---|---|
| 1 | 0.2 m | 0 kPa | 3.1 kPa | Destroyed house (dense flow $> 10$ kPa) |
| 2 | 0.2 m | 0 kPa | 2.6 kPa | Destroyed house (dense flow $> 10$ kPa) |
| 3 | 12.5 m | 71.9 kPa | 5.1 kPa | Large depostion (4.0 m) |
| 4 | 12.2 m | 73.1 kPa | 4.1 kPa | Large depostion (3.5 m) |
| 5 | 11.3 m | 72.5 kPa | 4.3 kPa | Large depostion (2.5 m) |
| 6 | 12.3 m | 13.1 kPa | 2.1 kPa | Large depostion (1.8 m) |
| 7 | 2 m | 0 kPa | 4.2 kPa | Damaged roof and balcony ($> 1$ kPa) |
| 8 | 0.5 m | 0 kPa | 2.3 kPa | Damaged balcony ($> 1$ kPa) |
| 9 | 5.3 m | 1.1 kPa | 2.3 kPa | Damaged roof ($> 1$ kPa) |
| 10 | 2 m | 0 kPa | 1.9 kPa | Damaged roof ($> 1$ kPa) |
| 11 | 0.4 m | 0 kPa | 0.9 kPa | Damaged roof and windows ($> 1$ kPa) |
| 12 | 11.2 m | 29.4 kPa | 1.7 kPa | Damaged windows ($> 1$ kPa) |
| 13 | 3.7 m | 0.8 kPa | 1.9 kPa | Damaged windows ($> 1$ kPa) |
| 14 | 0.7 m | 0 kPa | 2.5 kPa | Damaged windows ($> 1$ kPa) |
| 15 | 0.3 m | 0 kPa | 2 kPa | Damaged windows ($> 1$ kPa) |
| 16 | 0.2 m | 0 kPa | 2.8 kPa | Damaged windows ($> 1$ kPa) |
| 17 | 0.2 m | 0 kPa | 1.4 kPa | Damaged windows ($> 1$ kPa) |
| 18 | 6.8 m | 16.7 kPa | 1.3 kPa | Delimbed tree |
| 19 | 0.2 m | 0 kPa | 2.4 kPa | Delimbed tree |

the simulated suspension layer reaches all observed traces of the powder cloud without covering the region where no traces could be observed.

### 7.3.3 Real case example: The 2019 Eiskar Avalanche

On $15^{th}$ of January 2019, the Eiskar avalanche was released after intense snow falls and a quick temperature drop (Oesterle, 2019). The topography of the Eiskar avalanche differs drastically from the Wolfsgruben avalanche and thus provides a good supplement to that case. The avalanche was initiated by a collapsing slab on the right hand side of the avalanche path (looking in flow direction) and was falling on a larger snow field. From there, the avalanche slope continues with an inclination of approximately $25°$ for $1500$ m until reaching a flatter slope of $10°$. The dense flow avalanche ran $1000$ m on the flat slope and the powder flow exceeded the dense flow by another $500$ m, reaching the village of Ramsau. The powder cloud destroyed a wooden building, damaged a hotel and knocked over a bus. The dynamic pressure required for the damage was estimated at $1 - 3$ kPa. Areal pictures were taken after the event, which allowed to estimate the initial snow cover, the release area and deposition. The data was used to derive parameters for the snow cover function (Eq. (57)), $h_\Sigma(z_0) = 1.60$ m, $z_0 = 1275$ m, $\frac{\partial h_\Sigma}{\partial z} = 2 \cdot 10^{-3}$ to reach a snow cover thickness of approximately $2.7$ m at an elevation of $2200$ m (Oesterle, 2019). Other aspects of the simulation, such as the preparation of the terrain data match the simulation of the Wolfsgruben avalanche.



The first simulation (not shown) was conducted with the same parameters as for the Wolfsgruben avalanche. However, these
parameters lead to a severe underestimation of the powder cloud, running short by approximately 400 m. Simulations with the
model SamosAT (Sampl and Zwinger, 2004) showed similar results with the standard parameters (Oesterle, 2019). Therefore
the friction coefficients and the coefficient for the suspension feed were adjusted (see Tab. 3) to reach an appropriate runout
and dynamic pressure at the observed impacts.

**Figure 12.** Numerical simulation of the Eiskar avalanche with the two layer model. The first row (a-e) shows the height of the dense flow
layer, the second row (f-j) shows the height of the powder cloud layer. Each tick on the x- and y-axes corresponds to 500 m.



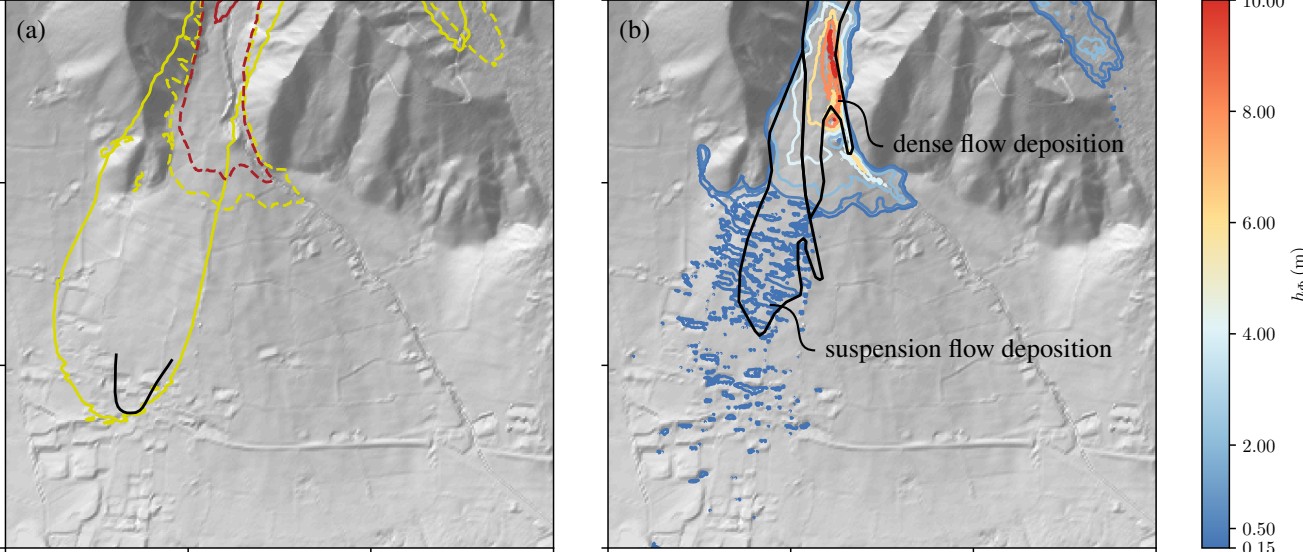

**Figure 13.** (a) The dynamic peak pressure of the suspension layer (solid lines) and the dense flow layer (dashed line). The yellow line marks the $1\,\mathrm{kPa}$ isoline and the red line the $10\,\mathrm{kPa}$ isoline. The black line marks the limit of the estimated $1\,\mathrm{kPa}$ isoline following observations. (b) The deposition of the avalanche (at $t = 200\,\mathrm{s}$). The black polygons mark regions of dense flow deposition (above) and powder cloud depostion (below).

Five timesteps of the simulation are shown in Fig. 12 in terms of the dense flow height $h_\Phi$ (a-e) and the suspension flow

height $h_\Pi$ (f-j). The collapsing slab (Fig. 12a) falls down the steep cliff onto a larger snow field where it can entrain additional snow. After around $30\,\mathrm{s}$ the avalanche reaches a second cliff and a powder cloud starts to emerge (Fig. 12b,g). The suspension layer keeps growing substantially in the slope section with an inclination of $25°$ (Fig. 12c, h) and starts to detach when reaching the flatter slope of $10°$ inclination. The suspension layer reaches the village of Ramsau after approximately $90\,\mathrm{s}$ (Fig. 12i, j) while the dense flow comes to a halt at the exit of the valley (Fig. 12d, e). Interestingly the dense flow is pushed towards the

left by terrain features at the exit of the valley while the suspension layer is widely unaffected by these small obstacles.

The corresponding zones of dynamic pressure are shown in Fig. 13a. The $1\,\mathrm{kPa}$ isoline of the suspension layer extends wide into the village. This fit was used as benchmark to determine the optimal model parameters and thus matches observations well. The final deposition of the model is shown in Fig. 13b and compared to the observed deposition. The observed deposition could be distinguish between suspension and dense flow depositions (Oesterle, 2019) and the same can be done in the numerical

model. The dense flow layer leaves behind up to $10\,\mathrm{m}$ thick deposits (to be corrected by a factor of $1/3$ to match the deposition density) with sharp edges, while the suspension generates deposits with $0.1-0.2\,\mathrm{m}$ thickness (to be corrected as well) that fade out gradually. Both, the position of the respective deposits as well as the rough shape match the observations.

Overall the model is able to reproduce the observed flow traces, from the dynamic pressure to the varying snow deposits in a single simulation. However, the model parameter had to be fitted to achieve these results. The friction parameters have

to be substantially lower than in the Wolfsgrube case and the coupling factor has to be an order of magnitude higher. This





indicated that either snow conditions were substantially different or that the model does not cover some substantially important processes.

## 8 Conclusions

This work provides an overview over the implementation of the granular dense flow model of Savage and Hutter (1989, 1991)
and the suspension flow model of Parker et al. (1986) into OpenFOAM. Further, the models have been combined by means of a novel coupling mechanism to provide a simple yet effective mixed snow avalanche model. These three models form the core of the OpenFOAM avalanche module. The module is accompanied by a new toolchain that substantially simplifies the practical application of the framework. The integration of geographic information system (GIS) file types into the OpenFOAM framework enables a simple and deep integration in existing workflows. Moreover, the dependencies on third party libraries for
GIS support were removed as they showed to be missing often on computational clusters. In comparison to the work of Rauter et al. (2018), the models and all tools are integrated into OpenFOAM to simplify installation. The physical models are highly modular. Tweaking and replacing specific empirical relation or process models is a core feature of the framework and highly encouraged.

The implementation of the suspension flow model of Parker et al. (1986) was verified by repeating published results, assuring
the absence of implementation errors. A novel two layer model was developed and evaluated with simple synthetic cases. The results are reasonable and follow the expectations set in the model. Further investigations have been conducted with two different real case avalanches. The reach of the dense flow layer and the suspension layer matched the observed runout in both cases with good accuracy, although a quantitative comparison was not conducted. The dense flow of the Wolfsgruben avalanche came short for approximately $20\,\mathrm{m}$, the impact pressure of the suspension flow is reasonable considering the observed damage.
Results for the Eiskar avalanche are similarly matching observations well if the parameters are fitted accordingly.

The good results are strongly linked to the parametrisation, which is highly uncertain due to the limited experience with mixed snow avalanche models in general and this model in particular. A wide variety of results can be achieved by tweaking the parameters of the model and substantial investigations will be required to find the appropriate parameters for the large number of semi-empirical relations embedded in the flow models. Substantially different parameters were required to yield
reasonable results in both cases, a well known problem in gravitational mass flow modelling (Scheidegger, 1973; Lucas et al., 2014). Further, snow properties and temperatures might have been substantially different between the two avalanche events. In this regard we see a strong opportunity to substantially improve the two layer model. Temperature has a strong influence on the particle diameter distribution in snow avalanches and will thus have a high effect on the mobility and the ability to generate suspension flows (Steinkogler et al., 2015a, b).
The dense flow runout and especially its dynamic pressure at a specific point are very sensitive to the parameters. This is related to the strong friction that rises rapidly in flat regions, where also the driving gravitational acceleration vanishes. The suspension cloud is less sensitive to such influences as the friction is lower and independent of the inclination and the basal pressure. Therefore the suspension runout is less sensitive to the parameters.



For practical applications we advice to use the existing guidelines for the dense flow parameters (e.g. Salm et al., 1990).
For snow avalanches with a high potential to generate powder snow clouds, we suggest to apply the suspension and coupling
parameters as used in the Eiskar case. It should be noted that the suspension model absorbs mass from the dense flow model,
which reduces the respective runout. Therefore it might be reasonable to simulate scenarios with less powder flow generation to
not underestimate the runout of the dense core. Finally, it should be kept in mind that the results of the model are accompanied
by a high amount of uncertainties and that they should be used accordingly. Nevertheless, the simulations presented here
recreate the processes of the events well and provide a considerable amount of additional information.

Generally, the model and the whole framework is aiming to be very flexible to provide researchers with a strong platform to
develop and evaluate novel friction, entrainment and coupling models. The introduced coupling model represents a reasonable
approach that yields promising results but there might be large opportunities for improvement. We hope that the framework
can provide a starting point for other researchers to develop new coupling mechanisms with better performance. Further, new
solvers can be implemented on basis of the framework, e.g. multiphase models for debris flows (e.g. Pudasaini, 2012; Kowalski
and McElwaine, 2013; Iverson and George, 2014) as done by Garcés et al. (2023) with `faDebrisFoam` or landslide tsunamis
(e.g George et al., 2017). The here presented toolchain and post-processing routines can be reused with these models and
additional pre- and postprocessing utilities can be added to enlarge the functionality of the whole framework.

*Code and data availability.* The code is available in the OpenFOAM avalanche repository at https://develop.openfoam.com/Community/
avalanche under the tag v2312. It is further included in the OpenFOAM-v2312 builds and releases (https://www.openfoam.com/news/
main-news/openfoam-v2312). The 1988 Wolfsgruben avalanche simulation and previous test and validation cases are included as a tuto-
rial in the repository. The code is licensed under GNU General Public License v3, test data is licensed under CC BY 3.0 by Amt der Tiroler
Landesregierung (AdTLR).

## Appendix A: Simplified computation of centrifugal forces

The basal pressure is computed following Eq. (19) in the (Savage and Hutter, 1989, 1991) model (Rauter and Tuković, 2018).
For the Parker et al. (1986) model we tried to achieve a simpler model that can also be combined with the empirical process
models in the powder cloud but still follows the general approach. Neglecting the small longitudinal pressure gradient term
and removing the indices marking the layer, Eq. (19) can be simplified to

$$h\,\rho\,\mathbf{g}_\mathrm{n} - \xi\,\rho\,\boldsymbol{\nabla}_\mathrm{n}^\Gamma \cdot (h\,\overline{\mathbf{u}}\,\overline{\mathbf{u}}) = -\mathbf{n}^\Gamma p_\Phi. \tag{A1}$$

We want to compare this equation to the following equation with an effective gravitational acceleration that contains the effects
of centrifugal forces,

$$h\,\rho\,\mathbf{g}_\mathrm{eff} = -\mathbf{n}^\Gamma p_\Phi. \tag{A2}$$



Setting $\mathbf{n}^\Gamma p_\Phi$ in Eqs. (A1) and (A2) equal to one another yields

$$h\,\rho\,\mathbf{g}_{\text{eff}} = h\,\rho\,\mathbf{g}_{\text{n}} - \xi\,\rho\,\boldsymbol{\nabla}_{\text{n}}^\Gamma \cdot (h\,\overline{\mathbf{u}}\,\overline{\mathbf{u}})\,. \tag{A3}$$

After approximating $\boldsymbol{\nabla}_{\text{n}}^\Gamma \cdot (h\,\overline{\mathbf{u}}\,\overline{\mathbf{u}})$ as $h\,\boldsymbol{\nabla}_{\text{n}}^\Gamma \cdot (\overline{\mathbf{u}}\,\overline{\mathbf{u}})$ and dividing by $h\,\rho$, we can write

$$\mathbf{g}_{\text{eff}} \approx \mathbf{g}_{\text{n}} - \xi\,\boldsymbol{\nabla}_{\text{n}}^\Gamma \cdot (\overline{\mathbf{u}}\,\overline{\mathbf{u}})\,. \tag{A4}$$

A further approximation neglects the shape factor $\xi$, finally leading to the effective gravitational acceleration as described by Eq. (31).

*Author contributions.* M.R.: Conceptualization, Methodology, Software, Writing. J.K.: Conceptualization, Validation, Writing.

*Competing interests.* No competing interests are present.

*Acknowledgements.* We thank Matthias Granig and Felix Oesterle (WLV) for providing, documentation, support and data for the real case examples.



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
