# Peer review of "OpenFOAM-avalanche 2312: Depth-integrated Models Beyond Dense Flow Avalanches"

_EGUsphere, 2024_

## Referee Comment (RC2)

[referee-annotated manuscript omitted]

---

## Editor Comment (EC2)

**Reviewer's Report on Manuscript egusphere-2024-210 "OpenFOAM-avalanche 2312: Depth-integrated Models Beyond Dense Flow Avalanches" by Matthias Rauter and Julia Kowalski**

Reviewed by Dieter Issler

April 23, 2024

**Content, novelty and suitability for GMD**

This manuscript presents a package of three depth-averaged open-source models for geophysical mass flows (GMFs): fa SavageHutterFoam for dense-flow avalanches (which could also be applied to rock or ice avalanches and with caveats to debris flows), faParkerFukushimaFoam for suspension flows without restriction to the Boussinesq regime, and faTwoLayerAvalancheFoam describing mixed snow avalanches (snow avalanches that develop a substantial suspension layer). The codes use the solvers of the OpenFOAM CFD library for finite-area meshes and are packaged with tightly integrated pre- and post-processing modules, among them a mesh generator. The first two models have been presented previously, at least to some degree, but their merged form, called faTwoLayerAvalancheFoam, is new. The geometrical setting, the governing equations, the closure assumptions and the numerical methods are explained in detail. In addition, the manuscript contains applications of faParkerFukushimaFoam to the idealized test cases used by Parker et al. (1986) to demonstrate the possibility of turbidity currents 'igniting' due to entrainment, and of faTwoLayerAvalancheFoam to a simple synthetic terrain for parameter studies and two well-documented powder-snow avalanche events in Austria.

The need to include the suspension layer in snow avalanche simulation has been felt acutely since the early days of avalanche hazard mapping in the 1970s and 1980s, and there have been numerous attempts at creating suitable numerical models for the purpose. Presently, mainly the Austrian two-layer model SAMOS-AT is being used in practice; it couples a depth-averaged (quasi-3D) flow model of the dense core to a depth-resolved (3D) model for the suspension layer. Consequently, computation times are substantial even on modern computers and considerable experience is needed to set up and assess simulations. The suspension layer can attain large heights and has a highly non-uniform velocity profile so that it is not obvious that depth-averaging is justified in this case. However, for practical purposes a reliable, easy-to-use and sufficiently quick code is of very high value; moreover, earlier codes like Eglit's two-layer model from the early 1980s and the quasi-2D code SL-1D (Issler, 1998) have shown in numerous applications that meaningful results can be obtained in this way.

It is thus clear that faTwoLayerAvalancheFoam is of great practical as well as scientific interest. With its focus on model description, numerical techniques and validation, the manuscript is very well suited for *Geoscientific Model Development* and should appeal to

many of its readers. As mentioned, the concept of a two-layer depth-averaged model is not new at all, and the governing equations of faTwoLayerAvalancheFoam are very similar to those of Eglit's (1982) model, SL-1D, MoT-PSA (Issler and Vicari, 2024) and—to some degree—RAMMS::EXTENDED (Bartelt et al., 2016). However, the authors' model contains several novel and interesting elements in the formulation of the closure relations, particularly for the suspension rate of small particles. Also, it applies a different numerical technique than all other existing models. In my opinion, the manuscript contains a sufficient amount of novel aspects as to merit publication.

**Presentation**

The manuscript is clearly structured and well written, striking a good balance between conciseness and detailed explanation of the important points. For most parts, the English is also good; in some passages, improvements are possible, and some suggestions are made in the annotated manuscript.

While the order of the sections is logical and pedagogically well justified, reading the paper feels somewhat tedious because of the length of Sec. 2, which explains the geometrical setting. I think many readers would appreciate it if the model content were presented earlier. One possible approach would be to relegate most of Sec. 2 to an appendix, as most of the material has been presented elsewhere by the first author.

The figures are pertinent and explain the points mentioned in the text well; they cover all issues that require a figure for explanation. A few minor improvements (zooming in in one case, visibility of lettering) are suggested in the annotated manuscript.

The authors have clearly made an effort at referencing copiously, yet this is one of two aspects of the manuscript that I am least happy with. See below under the heading *Major remarks* for my specific remarks.

**Major remarks**

**Curvature effects:** The authors formulate the governing equations of their system as what they call surface partial differential equations (SPDEs). Through depth-averaging, the 3D problem of flow over a surface $\Gamma$ is reduced to a 2D problem on $\Gamma$. This procedure is well understood and (fairly) straightforward if $\Gamma$ is a plane. For general curved $\Gamma$, the authors wish to circumvent the involved formalism of intrinsic differential geometry because it requires computation of the metric tensor and Christoffel symbols. Therefore, they work with the 3D velocity or momentum vector throughout, a fact they emphasized in an earlier paper by Rauter et al. and also ought to point out here. They use the projection operators onto the local tangent plane, $\nabla_s^\Gamma$, and onto the surface-normal direction, $\nabla_n^\Gamma$ to account for the geometry of $\Gamma$.

In their Eq. (7), $\partial \psi / \partial \mathbf{n}^\Gamma = \nabla \psi \cdot \mathbf{n}^\Gamma$, there is some ambiguity concerning the scope of the derivative operator, thus IMO it would be better to write $(\nabla \psi) \cdot \mathbf{n}^\Gamma$; this also applies to other equations throughout the manuscript. For vectors or tensors, the equation is correct both in a global Euclidean frame as well as in curvilinear coordinates if, e.g., a vector is written as $\mathbf{v} = v^i \mathbf{e}_i$ (using Einstein summation convention), but it is not valid for the component functions $v^i(\mathbf{x})$ alone if the basis vectors $\mathbf{e}_i$ vary spatially.

Equation (9) and the sentence embedding it appear to contain a misconception or poor wording: In the neighborhood of a point on a curved manifold, the manifold can be approximated by a paraboloid oriented along the normal vector at that point. This implies that curvature effects in the surface derivative of *scalar* functions are of order $(\Delta \mathbf{x})^2$ and vanish in the limit $\Delta \mathbf{x} \to \mathbf{0}$. In other words, the surface derivative of a scalar function defined in the embedding Euclidean space is equal to the tangential derivative. The same holds for the component functions $(v_x, v_y, v_z)$, relative to the global Euclidean basis, of a vector $\mathbf{v}$ in the embedding Euclidean space, and similarly for tensors.

Non-vanishing effects in depth-averaging over a curved surface arise from the deviation of $\det \mathbf{J}$ from 1, which is neglected in the authors' model. In a depth-averaged model on a curved surface, curvature effects besides $\det \mathbf{J} \neq 1$ arise because the velocity vector must be constrained to the tangent plane at every point. In models directly formulated on the curved surface, the basis vectors $\mathbf{e}_i$ vary spatially, implying that the component functions of a constant vector are not constant. This is captured by the covariant derivative, as described in all classical textbooks on differential geometry.

A second problem of Eq. (9) is that the dimensions do not match: the Gaussian curvature $K = \kappa_1 \kappa_2$, with $\kappa_{1,2}$ the two principal curvatures of $\Gamma$ at that point, has dimension $L^{-2}$. The authors cite (Dieter-Kissling et al., 2015) and (Tuković and Jasak, 2012) to support their claim. However, those authors do not use the Gaussian curvature $K$ but the *mean* curvature $H = (\kappa_1 + \kappa_2)/2$, which has the correct dimension $L^{-1}$.

Yet another problem of Eq. (9) is its failure to account for the tensorial nature of curvature. Take as an example a circular cylinder of radius $R$, the Gaussian curvature of which is identically 0 because the principal curvature in the axial direction is 0. Equation 9 then predicts that there are no centrifugal pseudo-forces, even if the flow moves normal to the cylinder axis along the circumference. In reality, there is a centripetal acceleration $u^2/R$. If the Gaussian curvature were replaced by the mean curvature, the centripetal acceleration in the azimuthal direction would no longer vanish but would be too small by a factor $1/2$ while there would be spurious centripetal acceleration in a flow along the axial direction.

I suspect that the authors overlooked some fundamental differences between Tuković and Jasak's (2012) interfacial flow model and GMF models: The thin-film model consists of a transport equation for the adsorbed surfactant mass, with the velocity determined by the bulk flows on either side of the interface; there is no separate momentum balance equation for the interface, only jump conditions for the interface-normal component of the stress tensor in terms of the mean interface curvature and the surface tension. This jump condition must sum the contribution from the surface tension over all directions. This is fundamentally different from the situation in a GMF, where the surface curvature only in the instantaneous flow direction determines the surface-normal stress.

Presumably, replacing the Gaussian curvature by the curvature in the flow direction will require non-negligible changes in the code because the principal curvatures in each cell and their orientation must be computed before the simulation so that the relevant curvature can be computed from them at each time step. Fortunately, this can be done quite efficiently using the first and second fundamental forms of

the surface, as in the codes MoT-Voellmy, MoT-muI and MoT-PSA. Nevertheless, fixing this and redoing all simulations with faTwoLayerAvalancheFoam represents a substantial effort and concerns a point that is not in the focus of this manuscript. In my opinion, it would be acceptable if the authors corrected the mathematical formulation in the paper and clearly stated that the simulations were run with an earlier version with an erroneous formulation of the curvature effects. It might be instructive to rerun one of the two real-world examples with the curvature term turned off to obtain an idea of the size of the effect. I suspect, however, that the Gaussian curvature is close to 0 along most of the thalwegs because the bottom of gullies resembles a cylinder.

**Referencing:** There are different opinions on whether one should give references for generally known facts. The authors chose to do so, but the selection of references appears haphazard in many cases. If there are dozens of standard textbooks explaining a specific point, choosing a lesser known and relatively recent one appears partial and is probably of little help to most readers. In such cases, my preference would be to say "as shown in all standard textbooks on fluid mechanics" or similar. However, the authors should let themselves be guided by the journals's and the editor's stance on this issue.

Selecting referenes becomes even more of a problem where the authors choose to reference journal papers that do deal with the aspect in question but do not represent a significant advance in this topic. In such cases, I think the seminal paper(s) ought to be cited. This situation arises in many places throughout the manuscript. Perhaps the most serious omission concerns the early Russian work on depth-averaged avalanche models from the 1D constant-density dense-flow models in the mid-1960s (Eglit, Grigorian, Yakimov and others) to Eglit's two-layer model of 1982, later developed further and tested extensively by Nazarov. That work preceded similar work in the West by roughly two decades, and it was accessible through several conference proceedings and translations yet largely ignored. The model described here does not differ from the Eglit–Yakimov model in its general set-up. Its innovations are mainly the 2D formulation (which was impractical in the early 1980s), a different numerical technique and different entrainment and suspension functions. This ought to be communicated clearly.

While I strongly dislike advertising my own work, two papers I have been involved in, from 1998 and 2024 respectively, are of some relevance here: (Issler, *Ann. Glaciol.* **26**, 1998) is a two-layer model that predates all two-layer models cited by the authors. It also is the basis for the code SL-1D, which was widely sold by SLF as part of the package AVAL-1D and still is in practical use in Switzerland today (despite its shortcomings). The very recently published paper (Vicari and Issler, MoT-PSA: a two-layer depth-averaged model for simulation of powder snow avalanches on three-dimensional terrain. *Ann. Glaciol.*, 2024, DOI 10.1017/aog.2024.10) is an extension of Eglit's model to 2D. It uses a simpler numerical technique than Eglit's or the authors', makes different choices for some of the closure relations and is from the ground up designed for use by practitioners with limited computational resources and little time to set up a complex software system. Otherwise, it is rather similar to the model described in this manuscript. I do not request the authors to cite these two papers, but they should at least be aware of their existence.

**Minor remarks**

Please see the attached manuscript for (a large number of) annotations on misprints, suggested changes of wording, minor remarks concerning the figures, etc. There are two points I wish to specially mention here:

- According to the README in the code repository, faSavageHutterFoam and faTwoLayerAvalancheFoam offer a variety of entrainment formulas. Such choice is very welcome because there is no good, generally accepted solution to the entrainment problem yet. However, IMO the authors have not chosen the models with the strongest physical foundation. In the manuscript, they explicitly present the model used by Fischer et al. (2015) in a calibration study. Assuming that the parameter $e_b$ is a constant, that model predicts the entrainment rate to increase linearly with the flow velocity if the bed shear stress is kept constant. It has been shown by several authors (Norem and Schieldrop, 1991; Fraccarollo and Capart, 2002; Issler and Pastor, 2011; Iverson, 2012; Issler, 2014; Iverson and Ouyang, 2015) that the entrainment rate must scale with the *inverse* of the velocity under such conditions because the eroded material must be accelerated to the flow velocity. Interestingly, this mechanism is incorporated in one of the two models proposed by Medina et al. (2007), but the authors chose to implement the other, which neglects the dynamics of the entrainment process. Equation (21) in the manuscript could be made more physical if one replaced $e_b$ by $e_b + \bar{u}^2/2$.

- The authors are very computer-savvy and therefore consider OpenFOAM-avalanche a simple, user-friendly system. Based on my experience both as a consultant and a researcher interacting with colleagues who do consulting work, I cannot quite agree with this assessment. While most practitioners are reasonably proficient with (their preferred) GIS, installing OpenFOAM and learning how to use it represents a higher hurdle than most practitioners are willing to surmount. I therefore suggest to tone down the respective statements.

**Recommendation to the editor**

While the second of my major criticisms can easily be remedied by rewriting parts of the introduction and adjusting the referencing in the rest of the paper, the first is more serious and more difficult to address. However, it is not critical for the main message of the paper and could be handled by pointing out that the simulations were run with an earlier code version. In view of this, I must leave it to the editor whether to ask the authors for a minor revision (but extensive in the small details) of their manuscript—preferably with another, quick round of review—or to request a major revision that fully addresses the curvature problem. Irrespective of this decision, I look forward to the eventual publication of this interesting and important manuscript.

Esentepe, 2024-04-23

Dieter Issler

---

## Author Response (AR1)

**Answers to reviewers on "OpenFOAM-avalanche 2312: Depth-integrated Models Beyond Dense Flow Avalanches"**

The original authors

June 22, 2024

The mayor concerns are addressed in the author replies. Here follow the answers to the comments that were embedded in the manuscript. Following the reviewers request, we just comment to the annotations that we did not implement. Comments are shown in blue, our answers are shown in black, line numbers refer the old version.

Line 1: This reads more like the beginning of the introduction or an abstract sent to a conference, where it is important to give background for non-specialists. Here, I would prefer a text that comes to the main points quickly and gives more technical information. For example, you do not mention that this concerns a two-layer model derived from the Savage–Hutter and Parker–Fukushima–Pantin models, that it includes entrainment and deposition and that you carry out comparisons with the PFP model and back-calculate two observed mixed avalanches.
The current version aligns with the journals guideline.

Line 22: A much earlier, conceptually simpler and less controversial example would be Eglit's (1982) model
We could not find this paper.

Line 31: It could be illunimating for many readers if you inserted the expressions for inertia and drag force before simplifying to the final result.
This would exceed the scope of the intrudction, since expressions for drag and interia have to be combined with assumptions of lenght and time scale.

Line 44: It is true that Sovilla et al. provide experimental evidence, but this fact has been known at least since the 1950s and probably even the 1930s. If I remember correctly, Hopfinger (1983) pointed this out as well.
We agree that Sovilla et al. are neither the only nor first to point this out. However, we

think this publication seems to be best suited to present the various regimes with many high quality figures.

Line 47: I would not consider the incompressibility as a characteristic feature but rather the collisional nature of the granular flow and the (perhaps) negligible role of interstitial air. For example, SL-1D (Issler, 1998), which is still being used by practitioners in Switzerand, treats the fluidized layer as a variable-density fluid and couples it with the variable-density suspension layer.

Dense granular flows can be simulated with incompressible models (see, e.g. Rauter, 2021), so it makes most sense for us to use such a model. We don't see the upside of a compressible model.

Figure 1: $\phi_s = 0.05$ corresponds to $\rho \approx 47 kg/m^3$. u = 100 m/s is close to the maximum of credible values, most PSAs move much more slowly. The maximum observed at Ryggfonn is around 60 m/s. However, the internal velocity a little above the dense-flow–suspension-layer interface could then reach about 100 m/s, as suggested by the lab experiments by Keller (1995). The height of the 1995 avalanche at Scex Rouge in Switzerland (Issler et al. 2020) reached about 400 m, as estimated from a photo, but in most cases the height remains below 100 m. At Vallée de la Sionne, the cloud height near the front was often about 20 m, with the very dilute wake attaining larger heights (at very low velocity). I would recommend replacing the single numbers in this figure by typical ranges, avoiding extremes. Perhaps mention that more extreme values have been reported.

Thanks a lot, we corrected some of them. It felt misleading adding full ranges for all properties, e.g. for height and velocity, which can potentially be 0.

Line 61: Your choice, but I find it fairly tiresome to read. Moreover, the choice of Greek letters $\Lambda$ and $\Sigma$ might be understandable if one thinks of German "Luft" and "Schneedecke", perhaps $\Phi$ stands for "Fließlawine", but what about $\Pi$ and what about non-German readers? Earlier authors used notation that is more convenient IMO.

I always thought of $\Lambda$ being similar to $A$ for air and $\Sigma$ standing for static. $\Phi$ (often the index for the granular phase in granular flow models) and $\Pi$ standing for the two moving grain layers. Hard to come up with something better and it is good to be in line with something readers might already know.

Line 81: A somewhat haphazard citation...

We cite the publication that we think is most useful for readers to follow up. We think this is a very good book on the Navier-Stokes Equations and CFD.

Line 86: Not a compelling citation either, since this has been discussed many times in the past century.

We cite the publication that we think is most useful for readers to follow up. This publication deals with different ways of describing multiple phases in CFD and seems to fit well here.

Line 93: Better: "tractable"

Not changed.

We cite the publication that we think is most useful for readers to follow up.

The theoretical description differs from the implementation. Details can be found in the cited paper.

Yes, exactly. Since there would be an excessive amount of terms, people neglect them in practice, often without explicitly mentioning that.

Not for depth integrated models.

People often think we speak of Navier–Stoke type models when describing the solution in 3D space (see also your remark above). This sentence should emphasise that we are still speaking about depth-integrated models.

We refer to existing works that discuss this aspect.

We wrote it like this in Rauter and Tukovic (2018). Now I think that notation is misleading and choose to not use it.

Explained a few lines below. $S_\Phi^\phi$ because only the grain volume is considered in this flux to be in line with the suspension flow model.

We think the short mentioning is sufficient.

NIS has the additional cohesion term, which makes a big difference, both in terms of requirements for the numerics and results. The used relation behaves very similarly to

the Voellmy relation, as we found out in Rauter et al. (2016). There are other friction models in the code to use Coulomb friction only.

Line 234: Note that this entrainment formula predicts an entrainment rate *growing* linearly with the flow velocity u if the bed shear stress is kept constant and $e_b$ is a constant. However, it has been shown by several authors that the entrainment rate must *diminish* as 1/u under these conditions because the eroded material must be accelerated to u. This could be fixed by adding $u^2/2$ to $e_b$.

We agree that the entrained snow masses have to be accelerated to the avalanche velocity. This is taken into account, at least to some extent, in the conservation equations: If the flow depth h grows due to entrainment, and the momentum hU is conserved, the velocity U decreases. We are not sure to which extent this covers your suggestion. The reasons for using this relation is given in the previous author response.

Line 263: It does hold when $\phi_\Pi < 10^4$, which is the case in the early and late phase of a PSA or if the PSA does not fully develop.

But not in general so we have to consider it.

Line 270: volume, mass

We don't see the need to mention volume, as volume conservation is only mass conservation with incompressibility.

Eq. (31): Since I do not agree with Eq. (9), I also object to this equation. Also, the nabla operator should be replaced by its projection along the bed-normal direction. Instead of n · [g $\nabla_n$ · (u u)], it should be written [...] · n because n · · f is not possible.

We removed Eq. (9) and reformulated the respective section, so also this equation should be fine now.

Line 328: Are there subaerial turbidity currents???

Considering the context of the paper, one could think of powder snow avalanches as subaerial turbidity currents. Anyway, we use subaquatic here to note that it is a classic "water and sediment" turbidity current.

Line 351: There is a rather important difference, however: Bartelt et al. (2016) describe a variable-density non-suspended flow, i.e., the lower layer can go back and forth between the dense and fluidized regimes.

Bartelt et al. (2016) seems closest from all models that have practical applications, at least at the time of writing this model and manuscript.

Eq. (46): This statement seems to imply that settling from the suspension layer creates a (often very thin) dense-flow layer because it cannot deposit directly on the snow cover. This would seem to make the notion of dense-flow runout rather tricky. This must be explained properly.

This is correct, the distinction is not tricky, as the depositions can be easily distinguished by their height, see simulation examples.

Line 386: This simplification is acceptable since you mention it. However, your argumentation does not take into account that the density at the botom of the suspension layer may well be ten times the density of air. What saves your day here is the tendency for the relative velocity between the layers to be smaller than the dense-flow velocity.
The density would still be small compared to the dense-flow, and, as you say, the velocity difference is expected to be small as well.

Line 374: I agree, but why do you account for it here but not in the suspension-flow model?
It is only applied to the cross-layer flux to limit unrealistically high phase fractions (even bigger than 100%) when the suspension layer is initiated by the dense flow. Such a situation is not possible in the suspension-flow model because the suspension is always picking up sediments itself and must therefore already contain some fluid to pick up the particles.

Line 392: There is one cross-layer coupling that you do not mention but is taken into account in many models since 1982: the weight of the powder-snow cloud increases the bed friction of the dense layer. This is a minor effect in the start but can be substantial under some conditions. It is OK to neglect it, but you ought to mention this.
We are not aware of such models. The powder cloud does not increase the friction of the dense-layer. The ice particles in the powder cloud are suspended so they do not contribute to effective pressure (e.g. Rauter 2020), which is the pressure that is relevant for granular bed friction.

Line 396: In the fluidized front of dry-snow avalanches developing a significant powder-snow part, small particles appear to be quite abundant, though. See (Issler et al., 1996, 2020) for more details.
Small particles have to be present in the dense core before they can form the powder cloud, we think that is clear.

Eq. (53): $I_0$ is often used as a parameter in the (I) rheology, so it might be wise to choose another index for this quantity.
The similarity is chosen on purpose.

Line 474: I do not understand the purpose of this.
Might be helpful for some readers.

Line 481: Better: (Rauter and Tuković, 2018) if you don't want the readers to contact you directly... Similarly for the other references in this sentence. We think the citation style is correct here but will be checked anyway by the typesetting team.

Line 487: in the sense of ensuring that the equations are solved correctly.
The chosen wording comes directly from the cited paper.

Line 524: Perhaps better: "chosen as"
Left as is.

Line 524: This is a monster avalanche...
The release volume is indeed very big but there is no entrainment, so the total avalanche size seems reasonable to us.

Line 537: This may be true for Austria but is not the case for the hazard zones in Switzerland elaborated by the reviewer between 2000 and 2006, using SL-1D and the default velocity profile derived from laboratory experiments by Keller (1996).

This refers to the equation for the dynamic pressure, not the governing equations. We could not find any guidelines or literature with a shape factor is used in the calculation of the dynamic pressure.

Line 555: Your assumptions for the initial conditions are well beyond what is observed, except perhaps in the Himalayas. Therefore it is hard to draw comparisons with real cases. In realistic situations, the powder-snow cloud often generates an extended "yellow" zone according to Swiss terminology (p < 3 kPa), larger PSAs may extend the "blue" zone (p < 3–30 kPa depending on avalanche frequency). I cannot remember having encountered a case where the cloud extended the "red" zone (p > 3–30 kPa depending on frequency). The avalanche released artificially in Vallée de la Sionne on 1999-02-25 exerted a pressure of about 50–70 kPa at the bunker on the oposite slope, but the culprit was not the suspension flow but the fluidized front, which you do not model explicitly.

The release volume is indeed very big but there is no entrainment, so the total avalanche size seems to be fine. Your description is in line with the two cases in the paper. In the avalanche intensity scale of Rapin are powder snow avalanches mentioned with > 10 kPa when they destroy forests. Also some of the observed snow avalanche damages require > 6 kPa, e.g. damaged roofs. Therefore it seems reasonable for the powder cloud to extend the 10kPa zone in some scenarios.

Table 3: Set to infinity?

Erosion was not active in this simulation as stated in the text. This can be achieved by setting the entrainment model to "NoEntrainment" in OpenFOAM. Here this is indicated by giving no parameter.

Line 594: Is there any information available on this matter? Based on my experience, this looks very plausible, but then the fluidized layer is not explicitly included in the model.

If $c_D$ is too small, the powder cloud completly outruns the dense core. We find this unrealisitic as we would see the dense core running after the powder cloud in experiments and real-case avalanches.

Fig. 8: What do these blobs signify?

They are not there when processed with ParaView, so it must be a post–processing issue. We left them in the image because we don't think manipulating result plots is ok. Notably the flow fields are a bit weird in the starting region (numerically vanishing powder cloud height, thus small fluxes change properties drastically).

Line 604: This value is, IMO, at the upper edge of the plausible range and would depend on when the density was measured and what the temperature was. One could of course argue that this is not the issue here, but the mass balance of the dense part plays a considerable role for its dynamics if there is a high degree of suspension. It would therefore be good idea to spen a sentence or two on the degree of entrainment (the same

Yes, it is on the upper edge, and so is the flow density. Anyway, it is convenient to get a round number. More precision would indicate an accuracy that's not there. This is just a rough value that explains the difference between the simulated and the documented deposition height which is roughly a factor of 3.

Line 607: This is not true, see e.g. (Issler et al., Geosci. 10, 2020).
Regarding "as no clear deposition pattern emerges from suspended flows". I am not sure if we looked at the correct publication, but in "The 2017 Rigopiano Avalanche—Dynamics Inferred from Field Observations" we didn't find any information on deposition patterns of the powdercloud, only other traces that also we use as proxies. Anyway, we think this holds, as people documenting these events can usually not delineate the poweder cloud extend with deposition. The deposition is further much more nuanced.

"...), stopping"
Left as is.

Confusing: one might understand this as saying that the dense-flow pressure exceeded 10 kPa, even though column 3 says p = 0 for the dense flow. Given the PSA pressure 3 kPa, please indicate the type of building. If these were masonry buildings that were utterly destroyed, the PSA pressure would likely be underestimated. These are very valuable constraints on the model. To withstand a pressure of 30 kPa, buildings usually must be built in reinforced concrete. Any information available? Not that trees often are broken or uprooted when hit by a PSA with this pressure. These are comparisons of simulated pressure with observed damages. So they do not match perfectly, as also described in the text.

I do not understand this statement very well. Presumably you mean that the model does not predict high pressures where the degree of observed damage precludes high pressure?
Yes, the area of "false positives" is small.

Fig 12: Is there observational evidence for this branch?
This branch is not documented in the official report, the aerial pictures start right below the branching. However, this branch is present in simulations with SamosAT, which is at least some evidence.

Line 659: I believe you mean that the parameters had to be chosen quite differently from the standard values used by WLV.
We fitted the parameter to achieve a match with the observations and they were different between the events. We didn't try to run it with values used by WLV.

Line 727: You do not mention another, usually even stronger approximation due to the height of the PSA cloud often approaching the curvature rasius of the terrain: $g_{eff}$ is not constant throughout the depth of the flow. I concede that this is difficult to capture in a depth-averaged model, but it ought to be mentioned.
This would be true if the flow is really surface aligned meaning that all streamlines are surface tangential. However, I don't even think that this is the case on strongly curved

terrain as eddies will form. Since this seems quite far stretched, we did not include it in the manuscript.